# ReXMoE: Reusing Experts with Minimal Overhead in Mixture-of-Experts

## Abstract

Mixture-of-Experts (MoE) architectures have emerged as a promising approach to scale Large Language Models (LLMs). MoE boosts the efficiency by activating a subset of experts per token. Recent works show that *fine-grained experts* substantially enriches the combinatorial flexibility of active experts and enhances model expressiveness. However, such a design is fundamentally limited by the *layer-local* routing mechanism: each layer is restricted to its own expert pool. This requires a careful trade-off between expert dimensionality and routing diversity given fixed parameter budgets. We describe ReXMoE, a novel MoE architecture that improves routing beyond the existing layer-local approaches by allowing routers to reuse experts across adjacent layers. ReXMoE decouples expert dimensionality from per-layer budgets, enabling richer expert combinations without sacrificing individual expert capacity or inflating overall parameters. To this end, we propose a new *progressive scaling routing* (PSR) strategy to gradually increase the candidate expert pool during training. As a result, ReXMoE improves both language modeling and downstream task performance. Extensive experiments on models ranging from 0.5B to 7B parameters across different architectures demonstrate that ReXMoE consistently improves performance under fixed architectural dimensions, confirming ReXMoE as new design paradigm for parameter-efficient and scalable MoE-based LLMs.

## 1 Introduction

Large Language Models (LLMs) have rapidly advanced in scale and capability, reaching hundreds of billions of parameters and demonstrating remarkable progress toward Artificial General Intelligence (AGI). Recent foundation models (Achiam et al., 2023; OpenAI, 2025; Meta AI, 2025; Guo et al., 2025; AI, 2025; Yang et al., 2025) have exhibited strong performance across complex tasks in multiple domains. This progress has been driven by massive investments in data and compute, but such growth also intensifies the tension between model capacity and development practicality. Given the substantial costs involved, Mixture-of-Experts (MoE) architectures have become an increasingly attractive alternative. By dynamically activating only a subset of specialized experts per input, MoEs can match or even exceed the performance of dense counterparts while significantly reducing inference-time computational demands (Shazeer et al., 2017; Lepikhin et al., 2020; Fedus et al., 2022; Du et al., 2022; Jiang et al., 2024; Liu et al., 2024a;b; Yang et al., 2025).

Comparing to the dense counterparts, a key characteristic of MoE architectures is the additional degrees of freedom when replacing the feed-forward networks with MoE blocks: the number of experts, the dimensionality of each expert, and the routing strategy. Recent studies on MoE scaling laws (Clark et al., 2022; Krajewski et al., 2024) highlight that model performance is constrained by trade-offs among these dimensions under a fixed parameter budget. In particular, the size of each expert and the number of experts form a critical axis: increasing the number of smaller experts enriches the space of expert combinations, whereas larger experts preserve stronger representational capacity but limit routing diversity. Such a trade-off is the core of the MoE architectural design.

In practice, recent works show trends toward adopting *finer-grained experts* in MoE design. For example, early Mixtral-of-Experts models (Jiang et al., 2024) employed 8 candidate experts per layer, whereas more recent models such as Qwen3 (Yang et al., 2025) series expand this to 128 experts, DeepSeek-V3 (Liu et al., 2024b) scales the design to 256 experts. From a combinatorial perspective,

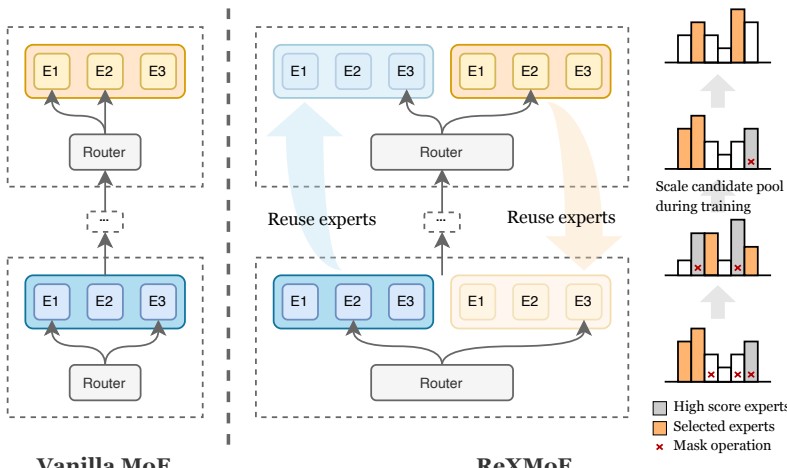

Figure 1: **Overview of REXMOE.** Compared to vanilla MoE, REXMOE enables more flexible expert combinations by reusing experts from adjacent layers. The only additional overhead comes from the router, which learns to route tokens to the expanded candidate pool. During training, REXMOE progressively scale the candidate pool by gradually reducing the number of masked experts until all experts are available. The mask operation sets corresponding gating score to 0.

fragmenting experts into smaller units substantially increases the number and diversity of possible routing combinations, thereby enhancing the expressiveness of MoE models and improving their ability to capture more specialized knowledge (Dai et al., 2024).

A key challenge in existing MoE designs lies in the *layer-local* routing mechanism, where each layer's router is restricted to its own expert pool. This constraint ties architectural choices to per-layer budgets and prevents more flexible balancing between the capacity of individual experts and the combinatorial flexibility of the expert pool. As a result, finer granularity comes at the cost of reduced representational capacity for each expert, since smaller experts correspond to reduced hidden dimensionality in their feed-forward networks. On the other hand, preserving expert dimensionality while simply increasing the number of experts inflates the overall parameter count. This fundamental challenge motivates us to explore new architectural directions that enrich expert combinations without reducing expert capacity or inflating model size.

To address such a challenge, we propose REXMOE, a novel approach to MoE architecture design that extends routing beyond the conventional *layer-local* boundary. By allowing routers to reuse experts across grouped adjacent layers, REXMOE decouples expert dimensionality from per-layer parameter budgets and introduces a new dimension in MoE design: models can realize richer expert combinations without sacrificing individual expert capacity or inflating the total parameter count. Furthermore, we present a Progressive Scaling Routing (PSR) strategy for training, which enhances the performance of models with reused expert pools. As illustrated in Figure 1, REXMOE reuses experts across adjacent layers with only negligible additional router parameters, while PSR gradually expands the candidate expert pool during training. Extensive experiments on models ranging from 0.5B to 7B parameters across different architectures show that REXMOE consistently improves performance under fixed dimensionality configurations. In addition, ablation studies highlight key design factors and confirm the practicality of our approach. Qualitative analysis further suggests that REXMOE enhances task-specific specialization. Together, these results establish REXMOE as an effective and scalable paradigm for advancing MoE-based LLMs.

In this paper, we make the following contributions:

- We design REXMOE, a method that breaks the limitation of *layer-local* routing in MoE architectures. By reusing experts across adjacent layers while adding only negligible router parameters, REXMOE significantly increases the flexibility of expert combinations while maintaining the representational capacity of each expert.

- We propose a *Progressive Scaling Routing* strategy in REXMOE, which gradually enlarges the candidate expert pool during training, thereby reducing language modeling loss and improving downstream task accuracy.
- We have conducted extensive experiments to demonstrate that REXMOE consistently improves both language modeling ability and downstream task performance across different model sizes and architectures, establishing REXMOE as a practical design paradigm for parameter-efficient and scalable MoE-based LLMs.

## 2 RELATED WORKS

**Mixture-of-Experts.** The strength of large models lies in their vast parameter counts, but this also brings the challenge of high computational cost. The Mixture-of-Experts (MoE) framework was introduced to decouple parameter size from per-token computation in large language models (LLMs) during both training and inference. In MoE-based transformer architectures (Vaswani et al., 2017), sparse MoE blocks (Shazeer et al., 2017; Fedus et al., 2022; Lepikhin et al., 2020) replace the Feed-Forward Networks (FFNs), improving efficiency while preserving strong performance. A notable example is Mixtral-of-Experts (Mixtral MoE) (Jiang et al., 2024), an open-source architecture that activates 2 experts from a pool of 8. Compared with dense models of similar computational cost, Mixtral delivers stronger performance across multiple downstream tasks. More recently, DeepSeek-MoE (Dai et al., 2024) adopts a fine-grained MoE design by dividing each FFN into smaller experts, enabling more flexible expert combinations without inflating the total parameter count. The open-source community has continued to advance this trend toward finer-grained experts. For instance, the Qwen3 series (Yang et al., 2025) employs 128 experts, while Kimi-K2 (AI, 2025) scales to 384 experts. Both demonstrate strong performance across diverse domains, reinforcing the idea that richer expert combinations often translate into better results. Overall, the success of these models highlights *finer-grained* design as a reliable and promising direction for future MoE architectures.

**Parameter Reusing in Transformers.** The standard Transformer constructs token representations using a stack of $L$ distinct layers, each consisting of a self-attention mechanism and a feed-forward network (FFN). Recent studies (Dehghani et al., 2018; Csordás et al., 2024; Bae et al., 2024) have explored reusing a shared set of weights across multiple layers, showing promising gains in parameter efficiency. Universal Transformers (Dehghani et al., 2018) replace the standard stack of unique layers with a single parameter-shared block that is applied recurrently, refining token representations in parallel. This combines the inductive bias of RNNs with the parallelization benefits of Transformers. MoEUT (Csordás et al., 2024) extends this idea by integrating the Mixture-of-Experts (MoE) paradigm into the recurrent Universal Transformer architecture, addressing its parameter–compute scaling bottleneck. This method increases model capacity while maintaining computational efficiency, making parameter sharing feasible for large-scale language modeling. Relaxed Recursive Transformers (Bae et al., 2024) further relax the strict layer-tying constraint by introducing depth-wise low-rank adaptation (LoRA) modules, improving performance while keeping the overall model compact. A related approach to parameter reuse in MoE blocks is WideNet (Xue et al., 2022), which derives its strategy from the perspective of reducing total parameters by recurrently reusing the weights of FFNs and self-attention blocks across all Transformer layers. Experiments on small-scale models for both CV and NLP tasks demonstrate its effectiveness, highlighting parameter sharing as a practical way to improve parameter efficiency.

## 3 METHOD

In this section, we first revisit the widely used TopK routing strategy for Mixture-of-Experts (MoE) models. We then introduce REXMOE, which enlarges the candidate expert pool by reusing experts across adjacent layers. To further improve performance when increasing the number of routed experts, we propose a Progressive Scaling Routing (PSR) strategy for training. An overview of REXMOE is shown in Figure 1.

### 3.1 REVIEW OF TOPK ROUTING MIXTURE-OF-EXPERTS

In a standard $L$-layer transformer-based Mixture-of-Experts (MoE) architecture, the Feed-Forward Network (FFN) blocks are replaced with MoE blocks, each comprising $N$ experts and a router. The

candidate expert pool of layer-$l$, $\mathcal{E}^l = \{E_1, E_2, \ldots, E_N\}$, consists of $N$ experts, each instantiated as an independent FFN. The router is responsible for assigning each input token to a subset of experts. Specifically, the router utilizes the gating network, which is parameterized by trainable weights, computes the probability distribution for the given input, then selects the corresponding experts according to its routing strategy. In TopK routing MoE, the output of the MoE block in $l$-th layer is computed as follows:

$$\mathbf{h}' = \sum_{i=1}^{N} \big( g_i \, E_i(\mathbf{h}) \big) \tag{1}$$

$$g_i = \begin{cases} s_i, & s_i \in \text{TopK}\left(\{s_j \mid 1 \leq j \leq N\}\right), \\ 0, & \text{otherwise}, \end{cases} \tag{2}$$

$$\mathbf{s} = \text{Softmax}(\mathbf{W} \cdot \mathbf{h}) \tag{3}$$

where $\mathbf{W} \in \mathbb{R}^{N \times d}$ is the weight of the gating network in the router, and $g_i$ is the gating score for expert-$i$. For brevity, we omit the self-attention and layer normalization in the above formulations.

## 3.2 EXPANDING CANDIDATE EXPERT POOL

To overcome the limitation of the *layer-local* routing mechanism, we expand the candidate expert pool by allowing the router to select from experts in *grouped* adjacent layers. Consider an $L$-layer MoE. Let $r$ denote the expert reuse frequency across layers, and let $\mathcal{E}^l$ represent the local candidate expert pool of the globally indexed $l$-th layer (with $l$ starting from 0). In REXMOE, the grouped candidate expert pool $\mathcal{U}_l$ for layer $l$ is defined as:

$$\mathcal{U}_l := \bigcup_{i \in G_l} \mathcal{E}^i, \quad G_l = \{\lfloor l/r \rfloor \cdot r + k \mid 0 \leq k < r\} \tag{4}$$

Here, group $G_l$ is formed by $r$ consecutive layers starting from the $\lfloor l/r \rfloor$-th layer. In this way, the candidate expert pool of each layer becomes $r$ times larger than in the vanilla setting. From the combinatorial perspective, the combination number is scaled from $C(N, k)$ to $C(rN, k)$. Now, the growth factor can be directly expressed as the ratio of the terms:

$$\text{ratio}(N, k, r) = \frac{C(rN, k)}{C(N, k)} = \prod_{i=0}^{k-1} \frac{rN - i}{N - i}, \tag{5}$$

which shows that enlarging the candidate pool significantly increases the number of possible expert combinations, offering a higher structural capacity ceiling. The computation of $l$-th layer's MoE block is then formulated as:

$$\mathbf{h}_l = \sum_{i=1}^{rN} \big( g_i \, U_i(\mathbf{a}_l) \big), \tag{6}$$

where $\mathbf{a}_l$ is the output of the attention block, $U_i \in \mathcal{U}_l$ is the $i$-th expert in the expanded pool, and $g_i$ is the gating score of the corresponding expert. By increasing $r$, the enlarged candidate pool enables more diverse expert combinations.

**Discussion.** A specific router configuration can restrict routing to only local experts; in this case, the model reduces to the vanilla MoE. This guaranties that our method always matches the baseline performance under the most constrained setting. When expert reuse is enabled, however, each MoE block includes a larger set of experts, which allows for more diverse expert combinations but can also lead to imbalanced routing patterns. Consequently, load imbalance becomes a critical bottleneck that not only limits generalization but also introduces challenges during training.

## 3.3 PROGRESSIVE SCALING ROUTING STRATEGY

Another key component of REXMOE is the Progressive Scaling Routing (PSR) strategy, which gradually increases the number of candidate experts during training. From an optimization perspective, enlarging the candidate pool also changes the gradient dynamics, since each expert receives

gradients aggregated from multiple layers within the same reuse group. Although this expansion improves the flexibility of expert combinations, the resulting gradient coupling limits the model's freedom to explore during training. This motivates the design of PSR, which balances the trade-off between gradient coupling and candidate pool expansion.

When reusing experts from $r$ layers in a TopK MoE with $N$ experts per layer, each router can access up to $rN$ candidates. Instead of training the router to select from all $rN$ candidates from the start, we adopt a progressive scheme: the number of available candidates begins at $N$ and is linearly expanded over the course of training. At iteration $t$, the candidate expert pool size $N_t$ is defined as:

$$
\mathrm{N}_t = \begin{cases}
N, & t \le t_s, \\
\left\lfloor (1 + \frac{(r-1)(t-t_s)}{t_e - t_s})N \right\rfloor, & t_s < t \le t_e, \\
rN, & t > t_e,
\end{cases}
\tag{7}
$$

where $t_s$ and $t_e$ specify the start and end iterations of the scaling schedule, respectively. At each iteration, we randomly mask $(rN - N_t)$ experts by setting their gating scores to zero before applying the TopK selection for each token. During the initial training stage ($t \le t_s$), each layer randomly selects $N$ experts from the expanded pool of $rN$. In this phase, the expected number of unique experts that are not simultaneously selected across layers is $rN(1 - 1/r)^{r-1}$. The gradients of these experts remain decoupled, which helps mitigate the limited exploration flexibility that often arises in the early stage of training. As training proceeds and the accessible expert pool is progressively enlarged, the model gradually learns to leverage the full structural potential enabled by the expanded combinatorial space, even under the constraints introduced by gradient coupling. This approach aligns with the principle of curriculum learning, guiding the model from early-stage gradient decoupling toward fully exploiting its expanded structural capacity.

### 3.4 PRACTICAL IMPACT ON PARALLEL STRATEGIES

The practical impact of REXMOE on large-scale distributed training is minimal. It remains fully compatible with established parallel strategies, including expert parallelism (EP) and pipeline parallelism (PP). For EP, the token dispatch distribution is only affected by the routing decisions, and the corresponding `all-to-all` communication volume matches that of a standard MoE system. Other EP optimizations remain applicable as well. For PP, placing reused blocks from the same reuse group across different PP stages introduces substantial parameter or gradient synchronization during training. It is therefore important to assign all reused blocks in a reuse group to the same PP stage to avoid cross-rank synchronization. With co-location applied, REXMOE behaves the same as a standard MoE system. If memory becomes the limiting factor and co-location is not feasible, increasing the EP size partitions experts across more devices, thereby reducing the per-GPU memory footprint. This helps enable co-location and supports larger reuse factors. With an appropriate PP–EP configuration, REXMOE can be used in both training and inference without adding communication overhead or causing new scaling bottlenecks.

## 4 EXPERIMENTS

### 4.1 EXPERIMENTAL SETUP

**Training environment.** All models are trained with Megatron-LM (Shoeybi et al., 2019), an open-source framework for large-scale language model training. We modified the `MoE Block` and `TopK Router` implementations to support cross-layer expert reuse and the Progressive Scaling Routing strategy during training. All models are pre-trained from scratch without instruction tuning, using the same hyperparameters across all runs. The sequence length is $4{,}096$ and the total batch size is $512$, resulting in a global batch size of 2M tokens. For optimization, we use AdamW (Loshchilov & Hutter, 2017) with $\beta_1 = 0.9$, $\beta_2 = 0.95$, weight decay $0.1$, and a gradient clipping ratio of $1.0$. The learning rate is scheduled to start at $3 \times 10^{-4}$ and decay to $3 \times 10^{-5}$ following a cosine schedule. We employ the sequence-level auxiliary loss and the bias-based router from (Wang et al., 2024). Further details are provided in Appendix B.2. All training jobs are conducted on 4 nodes, each equipped with $8\times$ NVIDIA Hopper GPUs.

Table 1: Base MoE architectures used in experiments. "MoE-0.5BA0.07B" denotes a MoE model with 0.5B total parameters and 0.07B active parameters per token. "SE" means "Shared Experts". This naming convention applies to all models.

| Model | Hidden Size | Intermediate Size | #Layers | Heads (Q / KV) | #Experts (Shared + Routed / Total) |
|---|---|---|---|---|---|
| MoE-0.5BA0.07B | 768 | 384 | 16 | 16 / 2 | 4 / 32 |
| MoE-0.5BA0.07B-SE | 768 | 384 | 16 | 16 / 2 | 1 + 3 / 32 |
| MoE-2.3BA0.3B | 512 | 744 | 32 | 16 / 2 | 8 / 64 |
| MoE-2.3BA0.3B-SE | 512 | 744 | 32 | 16 / 2 | 2 + 6 / 64 |
| MoE-7BA3B-SE | 2048 | 1408 | 32 | 16 / 4 | 2 + 6 / 64 |

**Model architecture.** We adopt the widely used Mixture-of-Experts (MoE) transformer architecture with consistent dimensionality settings across all ablation studies. The only differences lie in the router parameters under different reuse configurations. The architectural configurations are summarized in Table 1, where each model name specifies the number of activated and total parameters. The suffix "-SE" indicates that the architecture employs shared experts (Rajbhandari et al., 2022; Dai et al., 2024), and REX models follow the same naming convention. In addition, "-R$\{r\}$" denotes that experts are reused across $r$ layers.

**Training data.** We use the sample-100BT partition[1] from fineweb-edu dataset (Lozhkov et al., 2024; Penedo et al., 2024). The tokenizer is from LLaMA-2 (Touvron et al., 2023), with a vocabulary size of 32,000. Since the vocabulary is relatively small, the LLaMA-2 tokenizer does not achieve a high compression ratio. As a result, the processed 100B tokens cover around 87% of the original text. To ensure fair comparison, we fixed the data-parallel size and the shuffle seed, so that all experiments were trained on the same tokens in the same order, making the results directly comparable.

**Evaluation metrics.** We use lm-evaluation-harness (Gao et al., 2024) to evaluate performance on downstream tasks. Specifically, we report zero-shot accuracy on ARC-Easy (ARC-E) & ARC-Challenge (ARC-C) (Clark et al., 2018), BoolQ (Clark et al., 2019), HellaSwag (Zellers et al., 2019), LAMBADA (Paperno et al., 2016), LogiQA (Liu et al., 2021), OpenBookQA (Mihaylov et al., 2018), PIQA (Bisk et al., 2020), SciQ (Welbl et al., 2017), SIQA (Sap et al., 2019) and WinoGrande (Sakaguchi et al., 2021). For evaluation of the impact on inference speed after reusing experts from adjacent layers, we adapted REXMOE to vLLM (Kwon et al., 2023) and report the throughput (tokens per second) for prefill and decoding stages. Sampling is disabled in generation.

## 4.2 MAIN RESULTS

### 4.2.1 EVALUATION ON DOWNSTREAM TASKS

**Comparisons to vanilla MoEs.** We report the accuracy on downstream benchmarks in Table 2. The results show that the proposed REXMOE models consistently outperform vanilla MoE baselines across different model scales and benchmark tasks. Overall, REXMOE achieves stable improvements in both R2 and R4 configurations, with R4 often delivering the highest average accuracy. For example, compared to the base MoE-2.3BA0.3B, the R4 model attains the best results on tasks such as HellaSwag, LAMBADA, OpenBookQA, PIQA, and SIQA, raising the average score to 50.23%, which clearly surpasses the baseline's 49.15%. Similarly, under the "SE" setting, REXMOE-R4 outperforms the corresponding base MoE-2.3BA0.3B-SE. For smaller models in the MoE-0.5BA0.07B series, the advantage of REXMOE also remains consistent, where both R2 and R4 configurations yield notable gains in average accuracy over the baseline. More detailed task-wise accuracy trends during training can be found in Figure 6 and Figure 7 in the appendix. In summary, these results demonstrate that REXMOE consistently improves performance across different model scales and architectures, particularly on reasoning and knowledge-intensive tasks, highlighting its robustness, scalability, and general effectiveness.

---

[1]https://huggingface.co/datasets/HuggingFaceFW/fineweb-edu/viewer/sample-100BT

Table 2: Comparison between REXMOE and vanilla MoE models. All models are trained on 100B tokens. Task abbreviations: **Hella.** = HellaSwag, **LAMB.** = LAMBADA, **Lg.QA** = LogiQA, **Op.QA** = OpenBookQA, **Wino.** = WinoGrande. The best accuracy is highlighted in bold.

| Model | ARC-E | Hella. | LAMB. | Lg.QA | Op.QA | PIQA | SciQ | SIQA | Wino. | Avg.↑ |
|---|---|---|---|---|---|---|---|---|---|---|
| MoE-0.5BA0.07B | 50.67 | 38.38 | 32.37 | **28.42** | 31.00 | 65.29 | **71.20** | **38.84** | **53.04** | 45.47 |
| REX-0.5BA0.07B-R2 | 52.31 | 39.06 | **33.75** | 25.65 | 32.80 | 65.78 | 71.10 | 38.33 | 51.22 | 45.56 |
| REX-0.5BA0.07B-R4 | **53.91** | **39.46** | 32.76 | 25.35 | **32.80** | **66.81** | 71.00 | 38.38 | 52.17 | **45.85** |
| MoE-0.5BA0.07B-SE | 51.85 | 38.90 | 33.26 | 24.88 | 32.00 | 66.05 | 70.60 | **39.05** | 51.54 | 45.35 |
| REX-0.5BA0.07B-SE-R2 | 52.06 | 39.28 | 32.43 | 26.57 | **35.00** | 66.54 | 71.80 | 37.41 | **51.93** | 45.89 |
| REX-0.5BA0.07B-SE-R4 | **53.11** | **39.39** | **34.00** | **28.88** | 33.40 | **67.46** | **71.90** | 38.69 | 50.36 | **46.35** |
| MoE-2.3BA0.3B | 58.42 | 47.14 | 37.55 | 27.19 | 34.80 | 69.21 | 75.80 | 38.69 | **53.51** | 49.15 |
| REX-2.3BA0.3B-R2 | **61.32** | 46.84 | 37.20 | **28.57** | 35.00 | 69.48 | **76.50** | **39.61** | 52.33 | 49.65 |
| REX-2.3BA0.3B-R4 | 60.94 | **47.96** | **38.75** | 28.42 | **37.00** | **70.18** | 76.30 | 39.36 | 53.12 | **50.23** |
| MoE-2.3BA0.3B-SE | 58.42 | **48.79** | 38.13 | 25.35 | 37.00 | 69.53 | 75.00 | **40.28** | 52.17 | 49.41 |
| REX-2.3BA0.3B-SE-R2 | **59.09** | 47.99 | 38.54 | 27.34 | 37.60 | 69.48 | 74.20 | 39.56 | 52.72 | 49.61 |
| REX-2.3BA0.3B-SE-R4 | 58.71 | 48.59 | **39.01** | **28.26** | **39.00** | **70.67** | **76.10** | 39.66 | **52.80** | **50.31** |

Table 3: Comparisons between REXMOE and open-source models. We report results for models with equivalent total or activated parameters on selected language understanding benchmarks. Our method achieves competitive or superior performance across tasks. OLMoE is evaluated under the same environment as REXMOE, while scores for other models are taken from their original work.

| Model | #Act. Params | Data | ARC-E | Hella. | LAMB. | Lg.QA | PIQA | SciQ | Wino. |
|---|---|---|---|---|---|---|---|---|---|
| Llama2-7B (Touvron et al., 2023) | 7B/7B | 2T | **76.4** | **78.6** | **73.9** | 30.7 | 78.1 | 93.7 | 69.3 |
| MPT-7B-Base (Team, 2023) | 7B/7B | 1T | 67.3 | 76.1 | 70.3 | - | 79.9 | - | 68.3 |
| DeepSeekMoE-16B (Dai et al., 2024) | 3B/16B | 2T | 68.1 | 77.1 | - | - | **80.2** | - | **70.2** |
| LLaMA-MoE-8B (Zhu et al., 2024) | 3B/8B | - | 60.2 | 70.8 | 66.6 | 30.6 | 77.5 | 84.2 | 63.6 |
| OpenMoE-8B (Xue et al., 2024) | 2.1B/8B | 1T | 64.1 | 45.5 | - | - | 74.2 | - | 60.3 |
| OLMoE (Muennighoff et al., 2024) | 1B/7B | 5T | **76.4** | 77.0 | 73.3 | 29.7 | 80.0 | **95.1** | 68.8 |
| REX-7BA3B-SE-R3 | 3B/7B | 1T | 75.7 | 69.0 | 63.9 | **33.2** | 75.0 | 94.2 | 65.9 |

**Comparisons to LLMs with equivalent effective parameters**  We compare REXMOE with representative open-source dense and MoE models of similar total or activated parameter scales in Table 3. For a fair comparison, we scale the training data of REX-7BA3B-SE-R3 to 1T tokens sampled from fineweb-edu. The model exhibits well-balanced performance, achieving highest results on LogiQA, even when compared to Llama2-7B (Touvron et al., 2023), which uses more activated parameters and is trained on a larger corpus. Meanwhile, REXMOE remains highly competitive across the other benchmarks. These results demonstrate the effectiveness of REXMOE as model size and training data increase, highlighting its scalability for high performance.

### 4.2.2 IMPACT ON INFERENCE SPEED

We adapt REXMOE to the vLLM inference engine (Kwon et al., 2023) to evaluate the impact of expert reuse on practical applications. We fix the output length at 128 tokens and vary the input length to assess both prefill and decoding performance across different sequence lengths. The batch size is set to 1. The detailed results are shown in Figure 2. Although the computational overhead compared to vanilla MoE is negligible, REXMOE introduces a larger number of experts into each MoE block, which increases I/O operations during the prefill stage. As a result, the inference speed of the reuse scheme experiences a noticeable decline. As shown in Figure 2(a), a larger candidate expert pool leads to slower prefill speed, with the performance degradation being more pronounced when the input length is relatively short. Note that the absolute prefill latency of REXMOE is below 30 ms, and differences at this scale are negligible in practical use. As sequence length increases (e.g., 256), the relative slowdown becomes much less noticeable. More comprehensive results for sequence lengths up to 128k are provided in Table 12. Moreover, the prefill stage usually accounts for only a small portion of the total time, making the decoding stage of greater practical importance. As illustrated in Figure 2(b), REXMOE achieves comparable performance across different sequence lengths in

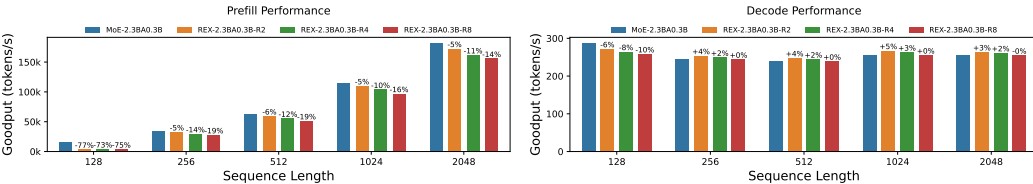

(a) Prefill goodput under different sequence length.  (b) Decode goodput under different sequence length.

Figure 2: Comparison of prefill and decode goodput between base MoE and REX models. Numbers above the bars indicate the relative speedup over the base MoE.

decoding stage. In practice, the model typically processes batches of requests. We further evaluate under varying batch sizes and observe that the degradation seen at sequence length 128 disappears. The detailed results are presented in Table 13, showing that expert reuse has no significant impact on practical deployment.

### 4.3 ABLATION STUDIES

#### 4.3.1 EFFECT OF EACH COMPONENT IN REXMOE

To demonstrate the effectiveness of each component in REXMOE, we provide comparative evaluations in Table 4. We utilize the MoE-2.3BA0.3B model as our baseline and set the expert reuse frequency across layers as 4. In addition to using the same average accuracy over the benchmarks reported in Table 2, we evaluate the validation perplexity (PPL) on WikiText (Merity et al., 2016). We find that the simple expansion of the experts' pool results in only a

Table 4: Average accuracy on benchmarks and PPL on WikiText (Merity et al., 2016).

| Model | Avg. Acc.↑ | PPL↓ |
|---|---|---|
| MoE-2.3BA0.3B | 49.15 | 21.19 |
| + Expert Reuse (4) | 49.28 | 21.12 |
| + PSR | **50.23** | **20.73** |

marginal improvement. Specifically, a $0.13\%$ increase in averaged accuracy on downstream tasks and $0.07$ in PPL. Furthermore, the incorporation of the PSR strategy yields a significant improvement in model performance by $1.05\%$ in the average accuracy and a drop in PPL of $0.46$. Comprehensive benchmarks results of each task can be found in Table 6. This behavior aligns with our expectation: due to gradient coupling, applying standard MoE training strategies directly to a reused architecture cannot fully realize its potential, and the resulting performance gains remain limited. With PSR, the capacity of the reused structure can be effectively unlocked, leading to significant improvements. These results demonstrate the effectiveness of expert reuse and the PSR strategy.

#### 4.3.2 EFFECT OF SCALING EXPERT REUSE GROUP SIZE

We investigate the impact of scaling the expert reuse group size in the 2.3B variant of REXMOE, where the reuse frequency ranges from 2 to 32 layers. As presented in Figure 3(a), we illustrate the performance trends on downstream tasks during training for different configurations. In the early training phase, both REX-R2 and REX-R4 underperform the baseline MoE model; however, they eventually surpass it as training progresses, with larger reuse group sizes generally leading to better performance. In contrast, REX-R16 and REX-R32 initially match or even exceed the baseline but later fall behind in the later stages of training. More detailed evaluation results can be found in Table 7 in the appendix. These results suggest that maintaining an appropriate balance in the number of reused layers is critical for sustaining high performance throughout training.

To investigate the cause of the performance decline as the number of reused layers increases, we reserve a validation set from the C4 corpus[2] to evaluate load balance. Following the MaxVio metric in (Wang et al., 2024), we adopt the Load Balance Violation (LBV) metric to quantify the degree of load imbalance in the MoE block. Specifically, the LBV of expert $i$ is computed as:

$$\text{LBV}_i = \frac{Load_i - \overline{Load_i}}{\overline{Load_i}} \tag{8}$$

---

[2]https://huggingface.co/datasets/allenai/c4

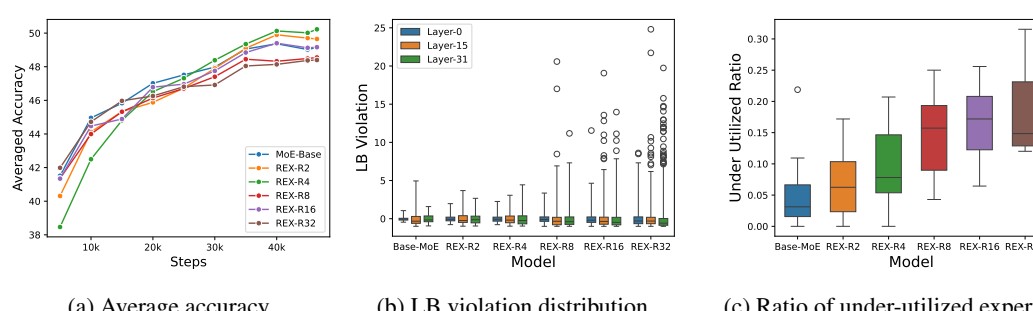

(a) Average accuracy.     (b) LB violation distribution.     (c) Ratio of under-utilized experts.

Figure 3: Average accuracy during training, distribution of load balance violations (degree of expert load imbalance), and distribution of under-utilized experts ratios (larger values indicate more inactive experts) under different cross-layer expert reuse sizes.

where $Load_i$ denotes the number of tokens assigned to expert $i$, and $\overline{Load_i}$ is the average expert load. Under perfect balance, $LBV_i$ equals $0$. As shown in Figure 3(b), larger $r$ values lead to more significant deviations among outliers in the distribution of $LBV_i$, indicating that the model tends to activate only a few experts and suffers from a collapse phenomenon during training.

Additionally, we analyze the distribution of under-utilized experts. In a Top-$k$ MoE, we define each expert's *active ratio* (utilized ratio) as the number of times it is activated divided by the total number of activations (i.e., #tokens $\times$ $k$). Under uniform activation, each expert should ideally have an active ratio of $k/N$. We therefore classify an expert as under-utilized if its active ratio falls below $t \times (k/N)$, where the threshold $t$ is set to $0.35$. In Figure 3(c), we present the distribution of under-utilized experts for each model across reuse groups. We observe that as the candidate pool further expands, more experts remain barely activated throughout training. This imbalance in expert utilization explains why the performance of REX-R16 and REX-R32 is lower than the baselines. While these results imply the load imbalance introduced by larger reuse factors, we note that this issue is fundamentally challenging to resolve. When $R \geq 8$, the total number of experts reaches 512–2048. In current large-scale practice, almost no production-level model employs more than 500 experts (e.g., DeepSeekV3 (Liu et al., 2024b) uses 256 experts, and Kimi-K2 (AI, 2025) uses 384). We argue that training with such extremely large expert pools and developing methods to mitigate the resulting load imbalance remains under-explored. We leave this direction to future work and hope these findings inspire further research.

### 4.3.3 EFFECT OF PROGRESSIVE SCALING ROUTING

We investigate an alternative Progressive Scaling Routing (PSR) strategy to validate the optimal configuration adopted in our main experiments. Specifically, we introduce PSR-Stepwise, which keeps the number of candidate experts fixed over certain training intervals. In contrast, the strategy described in Section 3.3 is referred to as PSR-Linear, as it provides a smoother and continuous expansion of the candidate expert pool. We use MoE-2.3BA0.3B as the baseline model and apply different PSR strategies to REX-R4. The corresponding training curves are shown in Figure 4. For both PSR-Stepwise and PSR-Linear, training starts

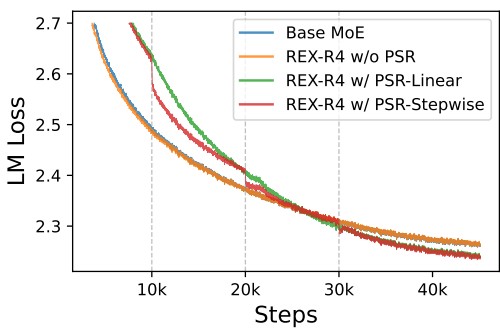

Figure 4: Loss curves for different strategies.

with 64 candidate experts, and scaling begins at step 10k. In PSR-Linear, the candidate pool is gradually increased to 256 by step 30k. In PSR-Stepwise, the candidate pool is set to 128, 192, and 256 at steps 10k, 20k, and 30k, respectively.

We present the training loss curves of these models in Figure 4. As shown in the figure, the loss curves of Base MoE and REX-R4, where PSR is not enabled, remain almost identical. When different PSR strategies are applied, model convergence is initially slowed. However, once the candidate

pool begins to expand, the models achieve lower loss than those trained without progressive scaling. Notably, PSR-Stepwise accelerates loss reduction during the mid-training phase.

As summarized in Table 5, both PSR strategies deliver clear performance improvements at final convergence, with detailed results provided in Table 6 in the appendix. Nevertheless, the final loss is comparable between the two strategies, while PSR-Linear achieves stronger overall performance on downstream tasks. Therefore, we adopt PSR-Linear to train all models.

Table 5: Average accuracy on benchmarks and PPL on WikiText (Merity et al., 2016).

| Model | Avg. Acc.↑ | PPL↓ |
|---|---|---|
| REX w/o PSR | 49.28 | 21.12 |
| REX w/ PSR-Stepwise | 49.59 | 20.76 |
| REX w/ PSR-Linear | **50.23** | **20.73** |

## 4.4 QUALITATIVE ANALYSIS

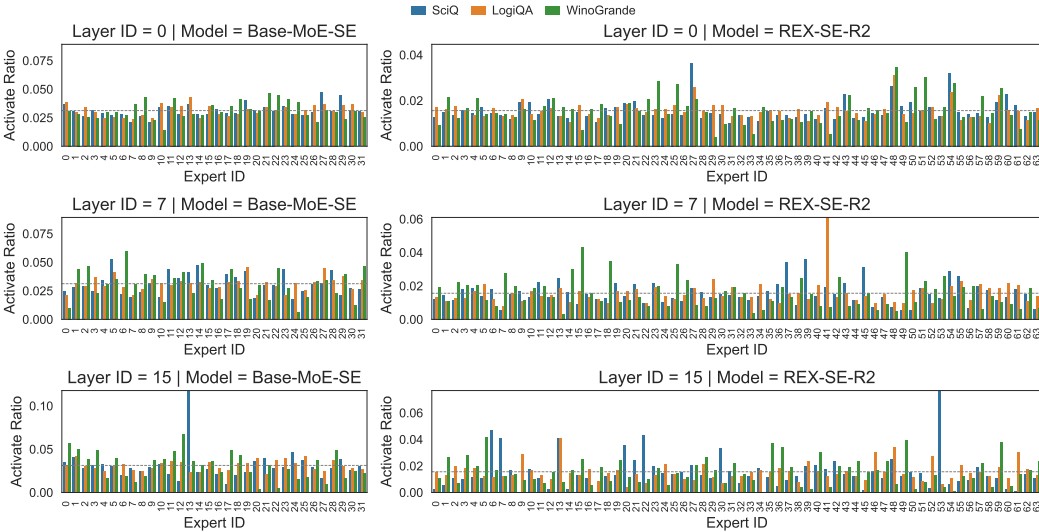

Figure 5: **Activate ratio of MoE-SE and REX-SE-R4 across layers in different tasks.** The gray dashed lines indicate uniform distribution. REXMOE shows stronger ability in task specialization.

In Figure 5, we present the expert activation ratios of Base-MoE-SE and REX-SE-R2 across layers 0, 7, and 15 on the SciQ, LogiQA, and WinoGrande tasks. The computation of the activation ratio follows Section 4.3.2. For Base-MoE-SE, the distribution of activated experts remains relatively uniform, with only minor variation across tasks. In contrast, REX-SE-R2 exhibits clear task-specific specialization. For instance, certain experts (e.g., Experts 25 and 49) are activated far more frequently for WinoGrande than for the other tasks, especially in Layers 7 and 15. Similar trends are observed in other tasks, as shown in Figure 8 and Figure 9 in the appendix. We also report the deviation of activation ratios among all layers in each model in Figure 10. These results suggest that the expanded expert pool of REX-SE-R2 allows for more effective task-specific allocation, encouraging the emergence of specialized experts and producing an ensemble-like effect in multi-task scenarios.

## 5 CONCLUSION

In this work, we present REXMOE, a novel MoE design paradigm that overcomes the limitation of *layer-local* routing. By allowing routers to reuse experts across grouped adjacent layers, REXMOE decouples expert dimensionality from per-layer budgets and substantially enlarges the candidate expert pool with only negligible router overhead. Combined with the Progressive Scaling Routing strategy, it further enhances training stability and performance. Extensive experiments across diverse architectures and model scales show that REXMOE consistently improves language modeling perplexity, downstream task accuracy, and the ability to learn task-specialized experts. Overall, these results establish REXMOE as a parameter-efficient and practically scalable paradigm for designing MoE-based LLMs.

REPRODUCIBILITY STATEMENT

We provide sufficient details for reproducing our key experiments. Training configurations are described in Section 4.1 and Section B.2, while the data processing pipeline is detailed in Section B.1.

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

# A    THE USE OF LARGE LANGUAGE MODELS (LLMS)

We acknowledge the use of Large Language Models (LLMs) to assist in writing and polishing this paper. Their role was limited to improving the clarity and readability of the manuscript; they were not involved in the design of the methodology or in the scientific analysis.

# B    ADDITIONAL EXPERIMENTS DETAILS

## B.1    DATA PROCESSING

We use the sample-100BT partition[3] of fineweb-edu (Lozhkov et al., 2024) for our main experiments. Each sample in the dataset is tokenized independently and then randomly concatenated into sequences of 4,096 tokens, which are used for training.

## B.2    HYPER-PARAMETERS AND PARALLELISM CONFIGURATIONS

We use the same hyper-parameters for all model training runs. The training sequence length is set to 4,096, and the global batch size is 512, resulting in a training batch size of 2M tokens. The base frequency for Rotary Positional Embedding (ROPE) (Su et al., 2024) is 10,000. For optimization, we use AdamW (Loshchilov & Hutter, 2017) with $\beta_1 = 0.9$, $\beta_2 = 0.95$, and a weight decay of 0.1, gradient clip ratio is 1.0. We adopt a warmup–cosine-decay learning rate scheduler, with an initial learning rate of $3 \times 10^{-4}$ that decays to $3 \times 10^{-5}$ by the end of training. The number of warmup steps is fixed at 100 for all experiments. When the number of routed experts exceeds 8, we enable Expert Parallelism (EP) with a parallelism size of 8 to accelerate training. No other parallelism strategies, such as Tensor Parallelism (TP) or Pipeline Parallelism (PP), are used in these runs. We globally fix the random seed to 42.

# C    ADDITIONAL EXPERIMENTAL RESULTS

## C.1    FULL EVALUATION RESULTS FOR DIFFERENT PSR VARIANTS

Table 6: Comparisons between base MoE and variants of REXMOE.

| Model | ARC-E | Hella. | LAMB. | Lg.QA | Op.QA | PIQA | SciQ | SIQA | Wino. | Avg.↑ |
|---|---|---|---|---|---|---|---|---|---|---|
| Base MoE | 58.42 | 47.14 | 37.55 | 27.19 | 34.80 | 69.21 | 75.80 | 38.69 | 53.51 | 49.15 |
| REX-R4 w/o PSR | 58.16 | 46.94 | 38.52 | 25.96 | 36.40 | 70.67 | 74.50 | **39.46** | 52.88 | 49.28 |
| REX-R4 w/ PSR-Stepwise | 60.65 | **48.25** | 37.67 | 27.04 | 34.40 | **70.84** | 74.60 | 39.10 | **53.75** | 49.59 |
| REX-R4 w/ PSR-Linear | **60.94** | 47.96 | **38.75** | **28.42** | **37.00** | 70.18 | **76.30** | 39.36 | 53.12 | **50.23** |

Complete evaluation results for different PSR variants are provided in Table 6, with the base model being MoE-2.3B-A0.3B.

## C.2    FULL EVALUATION RESULTS FOR DIFFERENT REUSE SIZES

Table 7: Comparisons between base MoE and REXMOE with different reuse sizes.

| Model | ARC-E | Hella. | LAMB. | Lg.QA | Op.QA | PIQA | SciQ | SIQA | Wino. | Avg.↑ |
|---|---|---|---|---|---|---|---|---|---|---|
| REX-R8 | 58.75 | 46.80 | 37.07 | 26.27 | 35.00 | 69.97 | 72.50 | 37.97 | 52.64 | 48.55 |
| REX-R16 | 58.59 | 46.79 | 38.48 | 27.80 | 35.40 | 70.02 | 72.20 | 39.36 | 53.83 | 49.16 |
| REX-R32 | 58.21 | 46.28 | 35.26 | 27.04 | 35.60 | 70.35 | 72.80 | 39.15 | 50.91 | 48.40 |

Complete evaluation results for different reuse sizes are provided in Table 7, with the base model being MoE-2.3B-A0.3B.

---

[3]https://huggingface.co/datasets/HuggingFaceFW/fineweb-edu/viewer/sample-100BT

## C.3 EVALUATION ON TOP2 MOE

We further apply REX to a Top2 MoE, with its architecture detailed in Table 8. The corresponding evaluation results are reported in Table 9.

Table 8: Architecture of Top2 MoE model used in the additional experiments.

| Model | Hidden Size | Intermediate Size | #Layers | Heads (Q / KV) | #Experts (Shared + Routed / Total) |
|---|---|---|---|---|---|
| MoE-0.5BA0.13B | 768 | 1536 | 16 | 16 / 2 | 2 / 8 |

Table 9: Comparisons between base MoE and REXMOE with different reuse sizes.

| Model | ARC-E | Hella. | LAMB. | Lg.QA | Op.QA | PIQA | SciQ | SIQA | Wino. | Avg.↑ |
|---|---|---|---|---|---|---|---|---|---|---|
| MoE-0.5BA0.13B | 51.85 | 38.70 | 33.09 | 27.34 | 33.00 | 66.05 | 67.40 | 38.54 | 51.38 | 45.26 |
| REX-0.5BA0.13B-R2 | 52.82 | 39.26 | 33.18 | 27.96 | 32.00 | 66.05 | 70.60 | 38.08 | 52.80 | 45.86 |
| REX-0.5BA0.13B-R4 | 51.94 | 39.34 | 32.25 | 27.04 | 32.80 | 65.56 | 70.60 | 38.69 | 50.51 | 45.41 |

## C.4 TASK-WISE ACCURACY

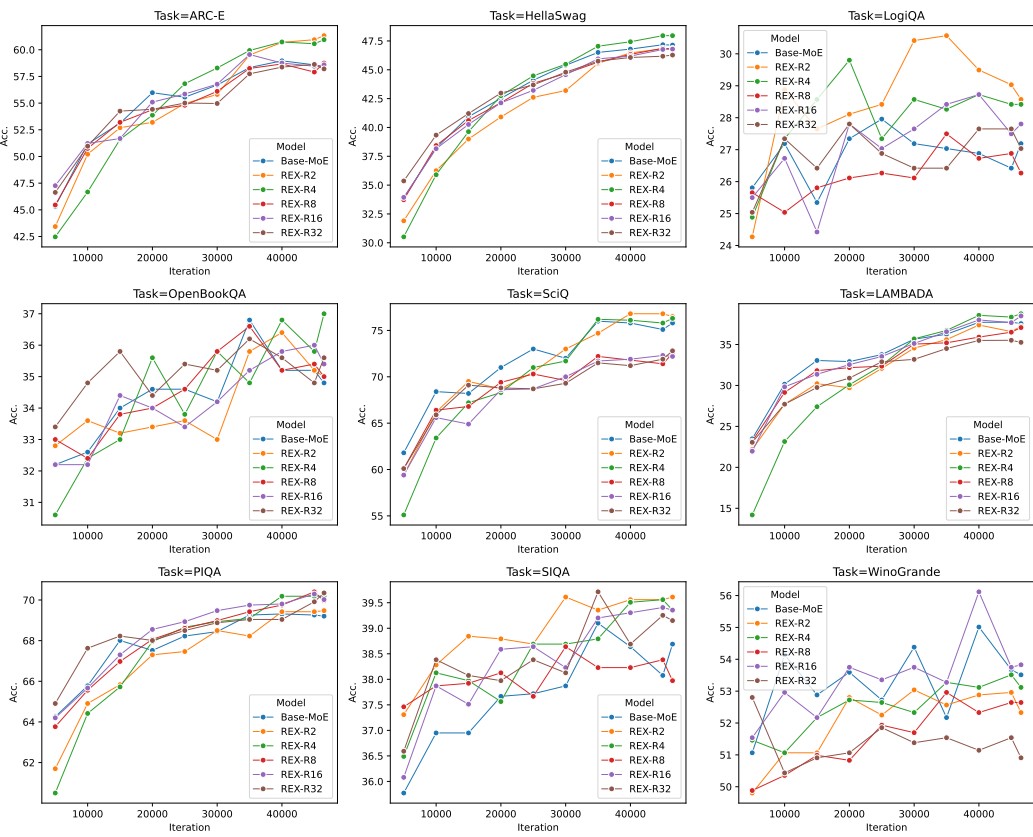

Figure 6: **Task-wise accuracy change as training progresses.** Base-MoE is MoE-2.3BA0.3B.

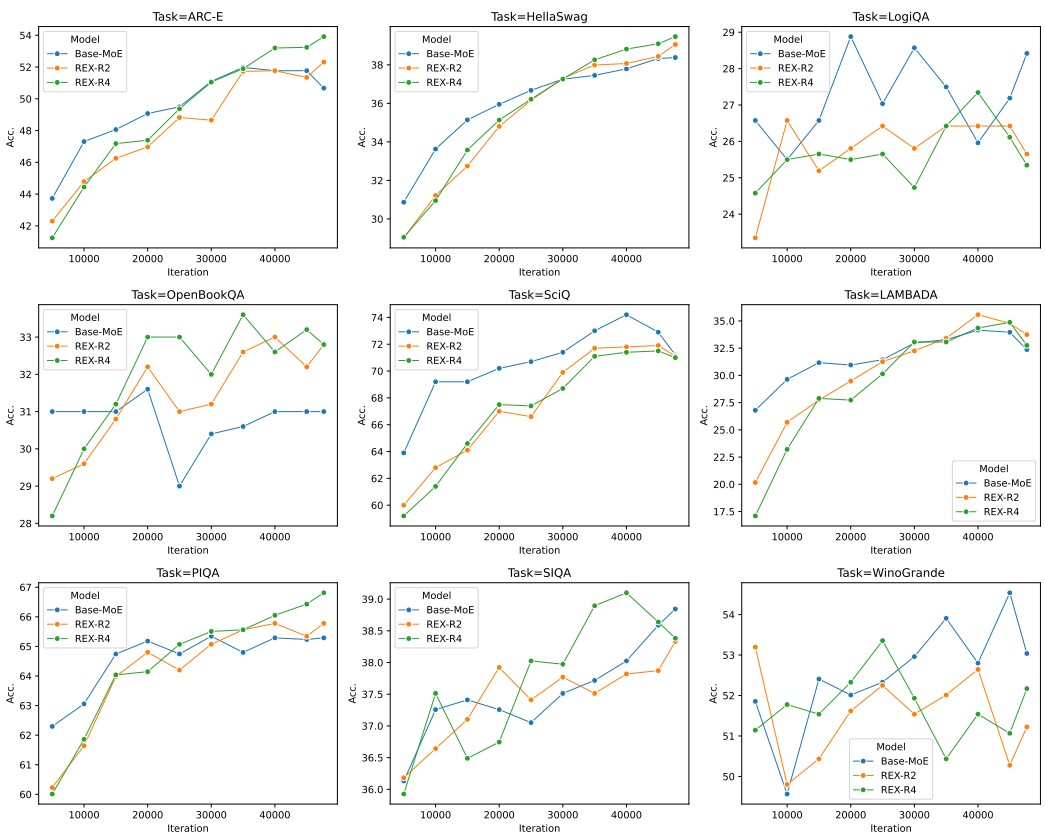

Figure 7: **Task-wise accuracy change as training progresses.** Base-MoE is MoE-0.5BA0.1B.

## C.5   TASK-WISE EXPERTS SELECTION VISUALIZATION

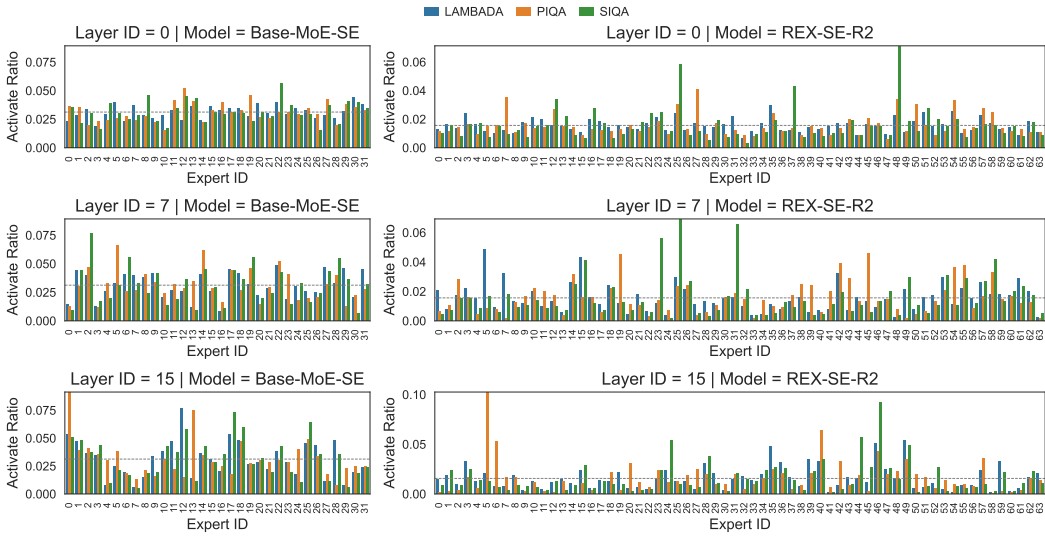

Figure 8: **Activate ratio of MoE-SE and REX-SE-R4 across layers in different tasks.** The gray dashed lines indicate uniform distribution.

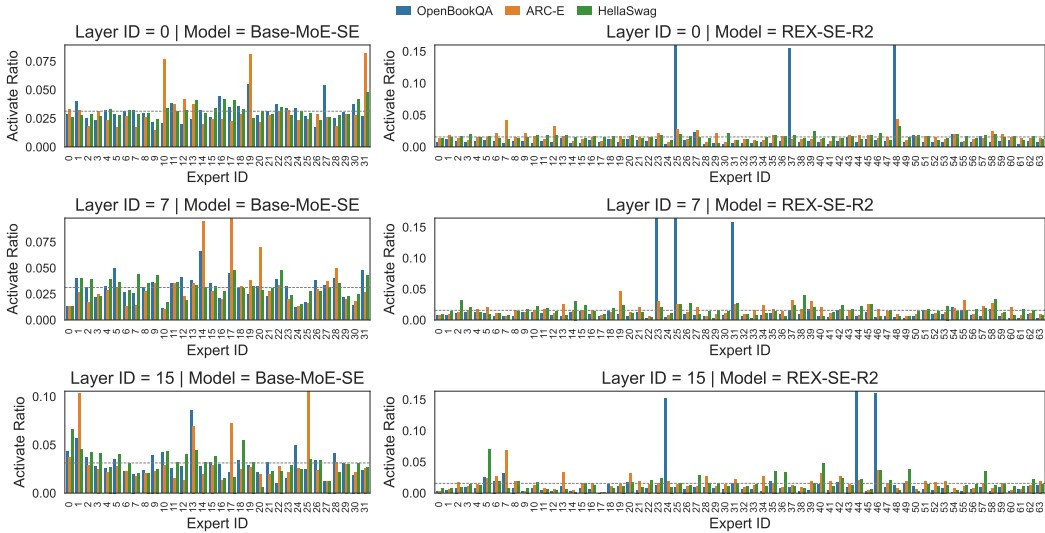

Figure 9: **Activate ratio of MoE-SE and REX-SE-R4 across layers in different tasks.** The gray dashed lines indicate uniform distribution.

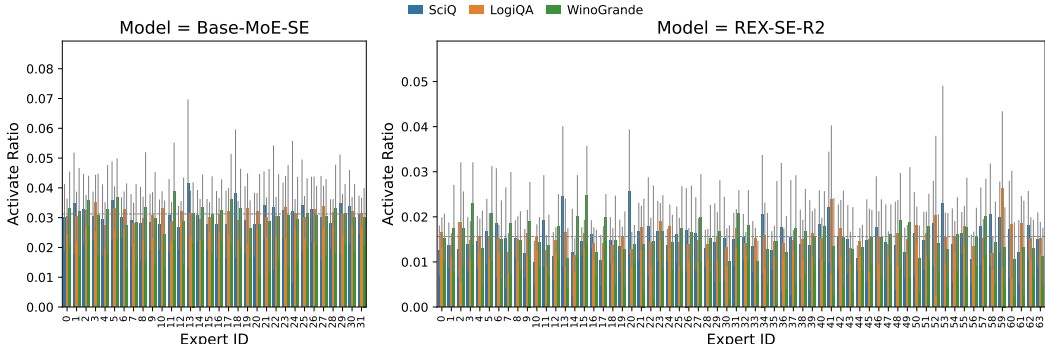

Figure 10: **Activate ratio distribution of MoE-SE and REX-SE-R4 in different tasks.** The gray dashed lines indicate uniform distribution.

## C.6 APPLYING PSR TO VANILLA MOE

To further isolate the effect of PSR, we apply it directly to a vanilla MoE model, MoE-2.3B A0.3B. As presented in Table 10, PSR degrades the performance of vanilla MoE. This is expected because PSR is designed specifically to counteract gradient coupling, which does not occur in vanilla MoE. Restricting the expert pool in this setting only reduces the flexibility of expert combinations, resulting in a performance decline.

Table 10: Performance comparison between base MoE and MoE+PSR.

| Model | ARC-E | Hella. | LAMB. | Lg.QA | Op.QA | PIQA | SciQ | SIQA | Wino. | Avg.↑ |
|---|---|---|---|---|---|---|---|---|---|---|
| MoE-2.3B A0.3B | 58.42 | 47.14 | 37.55 | 27.19 | 34.80 | 69.21 | 75.80 | 38.69 | 53.51 | **49.15** |
| + PSR | 58.04 | 45.64 | 36.10 | 27.65 | 33.60 | 68.72 | 74.60 | 38.69 | 52.41 | 48.38 |

## C.7  Parameter Efficiency Comparison Between ReXMoE and Inflating Parameters

To better illustrate the tradeoff between parameters and performance, we conducted an additional comparison in which we scaled the MoE-2.3B-A0.3B baseline to 3.5B parameters by simply adding more experts while keeping the number of active parameters fixed. Specifically, we added 32 additional experts, resulting in a total of 96 experts. The results are reported in Table 11.

Table 11: Performance comparison between ReXMoE and larger MoE.

| Model | ARC-E | Hella. | LAMB. | Lg.QA | Op.QA | PIQA | SciQ | SIQA | Wino. | Avg.↑ |
|---|---|---|---|---|---|---|---|---|---|---|
| MoE-2.3BA0.3B | 58.42 | 47.14 | 37.55 | 27.19 | 34.80 | 69.21 | 75.80 | 38.69 | 53.51 | 49.15 |
| MoE-3.5BA0.3B | 60.69 | 48.45 | 39.65 | 27.80 | 36.20 | 69.59 | 75.20 | 39.05 | 52.64 | 49.92 |
| ReX-2.3BA0.3B-R4 | 60.94 | 47.96 | 38.75 | 28.42 | 37.00 | 70.18 | 76.30 | 39.36 | 53.12 | **50.23** |

Increasing the vanilla model to 3.5B parameters, which raises the total parameter count by nearly 50%, yields only a 0.77% average improvement. In contrast, ReX-2.3B-A0.3B-R4 exceeds the performance of the 3.5B model **without increasing the parameter count**. These findings show that ReXMoE makes effective use of the additional structural capacity introduced by expert reuse and provides a clear advantage in parameter efficiency.

## C.8  Additional Evaluation on Inference Speed

Table 12: Prefill throughput and latency under larger sequence lengths. Base model is MoE-2.3BA0.3B. The values in parentheses represent the throughput ratio of ReX model compared to the base MoE.

| #Tokens | MoE-2.3BA0.3B | | ReX-2.3BA0.3B-R2 | |
|---|---|---|---|---|
| | Latency (ms) | Throughput | Latency (ms) | Throughput |
| 4,096 | 13.9 | 293,868.89 | 14.5 | 281,857.83 (95.91%) |
| 8,192 | 21.7 | 377,950.64 | 21.8 | 375,107.13 (99.25%) |
| 16,384 | 38.9 | 420,861.96 | 39.6 | 413,553.59 (98.26%) |
| 32,768 | 80.3 | 408,305.24 | 82.6 | 396,730.11 (97.17%) |
| 65,536 | 162.6 | 403,107.10 | 170.7 | 383,837.12 (95.22%) |
| 131,072 | 340.7 | 384,763.87 | 356.5 | 367,698.74 (95.56%) |

| #Tokens | ReX-2.3BA0.3B-R4 | | ReX-2.3BA0.3B-R8 | |
|---|---|---|---|---|
| | Latency (ms) | Throughput | Latency (ms) | Throughput |
| 4,096 | 15.1 | 271,765.00 (92.48%) | 15.6 | 262,995.96 (89.49%) |
| 8,192 | 22.8 | 359,752.24 (95.18%) | 22.6 | 361,892.12 (95.75%) |
| 16,384 | 39.8 | 411,431.91 (97.76%) | 40.9 | 400,309.30 (95.12%) |
| 32,768 | 80.7 | 405,821.25 (99.39%) | 82.9 | 395,358.85 (96.83%) |
| 65,536 | 162.4 | 403,486.21 (100.09%) | 166.8 | 392,812.21 (97.45%) |
| 131,072 | 345.6 | 379,278.06 (98.57%) | 358.1 | 366,035.86 (95.13%) |

Table 13: Prefill throughput under different batch sizes. The values in parentheses indicate the relative throughput ratio compared to the Base MoE.

| #batch | #tokens | MoE-2.3BA0.3B | REX-R2 | REX-R4 | REX-R8 |
|---|---|---|---|---|---|
| 4 | 128 | 4,268.8 | 4,274.7 (100.14%) | 4,256.0 (99.70%) | 4,062.3 (95.16%) |
| 4 | 256 | 18,635.5 | 17,777.0 (95.39%) | 16,418.0 (88.10%) | 14,811.4 (79.48%) |
| 4 | 512 | 35,207.4 | 33,388.7 (94.83%) | 31,386.3 (89.15%) | 28,506.0 (80.97%) |
| 4 | 1,024 | 68,443.5 | 65,190.8 (95.25%) | 62,040.4 (90.64%) | 57,692.8 (84.29%) |
| 4 | 2,048 | 90,164.7 | 87,695.1 (97.26%) | 83,484.9 (92.59%) | 75,432.5 (83.66%) |
| 16 | 128 | 7,974.5 | 7,540.6 (94.56%) | 7,553.7 (94.72%) | 6,871.9 (86.17%) |
| 16 | 256 | 11,646.8 | 10,811.5 (92.83%) | 12,964.0 (111.31%) | 13,129.6 (112.73%) |
| 16 | 512 | 20,082.1 | 20,157.3 (100.37%) | 19,310.2 (96.16%) | 17,524.2 (87.26%) |
| 16 | 1,024 | 27,734.1 | 27,069.8 (97.60%) | 25,874.7 (93.30%) | 27,636.1 (99.65%) |
| 16 | 2,048 | 34,163.9 | 34,681.1 (101.51%) | 33,375.0 (97.69%) | 33,660.8 (98.53%) |
| 64 | 128 | 3,396.9 | 2,616.2 (77.02%) | 2,264.7 (66.67%) | 2,414.7 (71.09%) |
| 64 | 256 | 5,665.4 | 5,522.8 (97.48%) | 5,362.1 (94.65%) | 5,460.0 (96.37%) |
| 64 | 512 | 7,633.2 | 7,499.9 (98.25%) | 7,487.0 (98.08%) | 7,674.6 (100.54%) |
| 64 | 1,024 | 9,114.4 | 9,214.5 (101.10%) | 9,026.5 (99.04%) | 9,001.5 (98.76%) |
| 64 | 2,048 | 9,847.1 | 9,889.4 (100.43%) | 9,928.1 (100.82%) | 9,684.4 (98.35%) |
| 256 | 128 | 1,011.2 | 939.3 (92.89%) | 925.2 (91.50%) | 930.1 (91.98%) |
| 256 | 256 | 1,901.3 | 1,935.4 (101.79%) | 1,838.4 (96.69%) | 1,506.3 (79.22%) |
| 256 | 512 | 2,288.9 | 2,268.7 (99.12%) | 2,290.7 (100.08%) | 2,253.3 (98.44%) |
| 256 | 1,024 | 2,305.0 | 2,342.1 (101.61%) | 2,299.6 (99.77%) | 2,404.7 (104.33%) |
| 256 | 2,048 | 2,552.9 | 2,555.4 (100.10%) | 2,553.7 (100.03%) | 2,542.8 (99.60%) |

