# OpenReview forum: "ReXMoE: Reusing Experts with Minimal Overhead in Mixture-of-Experts"
_ICLR.cc/2026/Conference — Submitted to ICLR 2026_

### Official Review · Reviewer_zRRq · 2025-10-30

**Soundness:** 3
**Presentation:** 2
**Contribution:** 2
**Rating:** 4
**Confidence:** 4

**Summary:**

This paper proposes REXMoE, a novel Mixture-of-Experts (MoE) paradigm that enables the reuse of experts across groups of adjacent layers. REXMoE extends routing beyond layer-local boundaries, allowing routers to leverage experts from other layers within the same group. Moreever, it introduces a curriculum learning–based strategy, Progressive Scaling Routing, to facilitate more scalable training. The authors pre-train REXMoE models at different scales and conduct comprehensive experiments to demonstrate their strong performance across various tasks.

**Strengths:**

- The idea of reusing adjacent layers of experts is novel and provides a straightforward method to enable a sparser MoE with more flexible routing mechanism while keeping the number of parameters fixed.

- The authors adapt REXMoE to vLLMs and provide a detailed analysis of inference speed, which enhances the practical applicability of REXMoE.

- The authors pre-train language models of different scales from scratch and conduct comprehensive experiments to demonstrate the effectiveness of REXMoE.

**Weaknesses:**

- Some parts of the paper’s presentation need further improvement (see Q1 & Q2).

- Some experimental results are not convincing enough to support the authors’ claims (see Q3, Q4, and Q5).

**Questions:**

---
Q1: What does "mask" means in Figure 1?

The term “mask” in Figure 1 is unclear. Could the authors provide a more detailed explanation of the mask operation?
Even after reading the full text, I find it difficult to fully understand how this operation works.

---
Q2: Is there a typo in Equation 4 and line 88?

From my understanding, the adjacent layers are partitioned into different groups.
Accordingly, Equation 4 should be reformulated as:

$\mathcal{G} = \{\lfloor l/r \rfloor \cdot r + k \mid 1 \le k \le r\}$，

The current form of Equation 4 is somewhat unclear and may cause confusion.
Hope the authors can clarify that.

---
Q3: Does REXMoE achieve its superior model performance by accelerating the convergence of pre-training?

Could the authors pre-train a smaller REXMoE model with more tokens (e.g., 350B or 500B) and report results similar to those in Table 2?
Conducting more extensive pre-training would provide stronger evidence to convincingly demonstrate REXMoE’s effeciveness.

---
Q4: Concerns regarding the results presented in Table 3

(1) It's unclear why certain results are missing, given that these open-sourced models can be readily evaluated using tools like LM Eval.
Completing the table will provide a more comprehensive and convincing comparison.

(2) The claim of competitiveness between REXMoE and Llama 2 (as stated in the caption) is currently unconvincing to me.

(3) Can authors include results for other competitive open-source models, such as OLMoE [1].
Additionally, to better support claims of superiority in reasoning and knowledge-intensive tasks, the authors can conduct further evaluations on benchmarks like GSM8K and MMLU.

[1] OLMoE: Open Mixture-of-Experts Language Models

---
Q5: Can the the authors provide an inference speed comparison for the 7B REXMoE?

On one hand, I am curious about how the inference efficiency of larger-scale REXMoE compares to that of standard MoE models.

On the other hand, when r is larger, it becomes difficult to include r layers of experts within a single GPU (worker), and could further complicate the  pipeline parallelism.
I hope the authors can clarify how this challenge can be addressed.

---

> ### Author Response · Authors · 2025-11-23
> **Response to Reviewer zRRq (1/3)**
>
> We thank the reviewer for their time and valuable feedback. We have carefully considered all comments and have revised the manuscript accordingly with the modifications highlighted in blue. Below, we provide detailed responses to each question.
>
> ---
>
> > Q1: What does "mask" means in Figure 1?
> >
> > The term “mask” in Figure 1 is unclear. Could the authors provide a more detailed explanation of the mask operation? Even after reading the full text, I find it difficult to fully understand how this operation works.
>
> The mask operation sets the gating score of a particular expert to **zero**, effectively removing that expert from the routing candidates for the current token. We have clarified this operation in the revised manuscript; please refer to line 76 and line 231.
>
> ---
>
> > Q2: Is there a typo in Equation 4 and line 88?
>
> Thank you for pointing this out. There is indeed a typo in Equation 4: the term should be multiplied by $r$. We have also improved the notation for clarity.
>
> The grouped candidate expert pool $\mathcal{U}_l$ for layer $l$ is now defined as:
>
> $$
> \\mathcal{U}_l := \\underset{i \\in {G_l}}{\\bigcup}  \\mathcal{E}^{i},
> $$
>
> where
> $$
> G_{l} = \\{\lfloor l / r \\rfloor \\cdot r +k \\mid 0 \\leq k < r \\}
> $$
>
> Here, group $G_l$ is formed by $r$ consecutive layers starting from the $\\lfloor l / r \\bigr\\rfloor$-th layer.
>
> Regarding the potential typo in line 88: we checked carefully but could not locate the issue. If possible, could you kindly clarify which part seems incorrect?
>
> ---
>
> > Q3: Does REXMoE achieve its superior model performance by accelerating the convergence of pre-training?
> >
> > Could the authors pre-train a smaller REXMoE model with more tokens (e.g., 350B or 500B) and report results similar to those in Table 2? Conducting more extensive pre-training would provide stronger evidence to convincingly demonstrate REXMoE’s effeciveness.
>
> We appreciate the reviewer for raising this key question. Our clarification is as follows:
>
> In the early stage of training, PSR slightly slows convergence due to the restricted candidate pool. However, once the pool expands, ReXMoE consistently achieves lower loss than vanilla MoE. **Figure 4** illustrates this comparison. This shows that ReXMoE’s improvement stems from the enlarged candidate expert pool rather than accelerated convergence.
>
> Additionally, following the reviewer’s suggestion, we pre-trained a **smaller 155MA46M MoE model** on **350B tokens**, and compared it with a vanilla MoE counterpart. The results are shown below:
>
> |                 | ARC-E | HellaSwag | LAMBADA | LogiQA | OpenbookQA | PiQA  | SciQ  | SiQA  | WinoGrande | Avg.  |
> | --------------- | ----- | --------- | ------- | ------ | ---------- | ----- | ----- | ----- | ---------- | ----- |
> | MoE-155MA46M    | 44.99 | 36.07     | 30.14   | 26.57  | 31.00      | 64.42 | 61.70 | 38.08 | 51.07      | 42.67 |
> | ReX-155MA46M-R4 | 46.57 | 36.74     | 29.78   | 26.57  | 32.60      | 65.07 | 64.80 | 38.97 | 51.22      | 43.59 |
>
> The R4 model achieves a **0.92% average improvement**, confirming that ReXMoE remains consistently better under a significantly larger training budget. These findings strengthen our conclusion that the performance gains are not attributed to faster convergence but to the improved structural capacity enabled by expert reuse and PSR.
>
> We thank the reviewer again for this insightful suggestion, which helped further validate the robustness of our method.

---

> ### Author Response · Authors · 2025-11-23
> **Response to Reviewer zRRq (2/3)**
>
> > Q4: Concerns regarding the results presented in Table 3
> >
> > (1) It's unclear why certain results are missing, given that these open-sourced models can be readily evaluated using tools like LM Eval. Completing the table will provide a more comprehensive and convincing comparison.
> >
> > (2) The claim of competitiveness between REXMoE and Llama 2 (as stated in the caption) is currently unconvincing to me.
> >
> > (3) Can authors include results for other competitive open-source models, such as OLMoE [1]. Additionally, to better support claims of superiority in reasoning and knowledge-intensive tasks, the authors can conduct further evaluations on benchmarks like GSM8K and MMLU.
> >
> > [1] OLMoE: Open Mixture-of-Experts Language Models
>
> We thank the reviewer for these insightful questions. Below, we address each concern in detail.
>
> **(1) Missing results in comparison with open-source models**
>
> We would first like to clarify the rationale behind selecting the models included in Table 3. Our goal was to compare REXMoE with open-source models that:
> - have similar parameter or activation budgets, and
> - are trained on comparable amounts of data,
> in order to provide a meaningful reference for the scaling potential of ReXMoE.
>
> In all our experiments, we evaluated models under **exactly the same environment** (same versions of lm-evaluation-harness, transformers, flash attn, etc.) to ensure stability and reproducibility of comparisons. However, for several open-source models, evaluation under this unified environment produced unreliable or clearly incorrect results:
>
> - **MPT-7B** and **OpenMoE-8B** depend on older PyTorch versions and outdated custom kernels. After downgrading the environment and applying several workarounds, inference could run, but the outputs were incorrect, and benchmark scores became unrealistically low.
> - **DeepSeekMoE-16B** can be evaluated in our environment, but its results significantly diverge from those reported in the original paper. For reference:
>
> | Model                            | ARC-E | Hella. | LAMB. | Lg.QA | PIQA | SciQ | Wino. |
> | -------------------------------- | ----- | ------ | ----- | ----- | ---- | ---- | ----- |
> | DeepSeekMoE-16B (original)       | 68.1  | 77.1   | –     | –     | 80.2 | –    | 70.2  |
> | DeepSeekMoE-16B (ours evaluated) | 73.0  | 72.3   | 73.6  | 29.4  | 75.7 | 90.2 | 70.2  |
>
> The discrepancies on ARC-E, HellaSwag and PIQA make the evaluation unreliable in a strictly controlled setting.
>
> For this reason, we opted to report the **published results** from their original work rather than mixing them with inconsistent in-house evaluations. We have updated the caption of Table 3 accordingly to clarify this choice.
>
> **(2) On the claim of competitiveness between REXMoE and Llama 2**
>
> We agree that a fully fair comparison is difficult because these models are trained on different corpus and follow distinct training recipes. Our intention in Table 3 is to show that **ReXMoE remains competitive at larger scales** (7B parameters, 1T tokens), not to claim absolute superiority.
>
> Regarding the comparison with Llama 2 specifically:
> Our model is trained on **FineWeb-Edu** [1], a high-quality web dataset. This corpus is considered cleaner and more curated than portions of Llama 2's training data, which may explain why ReXMoE shows relatively stronger performance on some tasks.
>
> **(3) Comparison with OLMoE and results on GSM8K & MMLU**
>
> Thank you for the suggestion. We have now added OLMoE to the comparison. We acknowledge that OLMoE performs notably better than REXMoE across many tasks; however, it is important to note that OLMoE is trained on **5T tokens of high-quality data** and uses a more carefully tuned annealing schedule. In contrast, our results reflect performance from a single pre-training run under a 1T-token budget.
>
> We also include new evaluations on **GSM8K** and **MMLU**, as requested. The results are summarized below:
>
> | Model           | GSM8K    | MMLU     |
> | --------------- | -------- | -------- |
> | Llama2-7B       | 14.6     | 45.3     |
> | MPT-7B-Base     | –        | 29.6     |
> | DeepSeekMoE-16B | **18.8** | 45.0     |
> | LLaMA-MoE-8B    | –        | 26.8     |
> | OpenMoE-8B      | –        | 26.2     |
> | OLMoE           | 3.0      | **54.1** |
> | REX-7BA3B-SE-R3 | 2.3      | 32.6     |
>
> These results highlight two observations:
> - Most models trained solely on **public web data** perform poorly on GSM8K due to lack of  exposure to math data.
> - ReXMoE’s MMLU score is above random guessing (~25%) and shows a upward trend, but likely requires additional more training to reach competitive levels.
>
> We appreciate the reviewer’s suggestions, which have helped improve the completeness and clarity of our comparisons.
>
>
> [1] https://huggingface.co/datasets/HuggingFaceFW/fineweb-edu

---

> ### Author Response · Authors · 2025-11-23
> **Response to Reviewer zRRq (3/3)**
>
> > Q5: Can the the authors provide an inference speed comparison for the 7B REXMoE?
> >
> > On one hand, I am curious about how the inference efficiency of larger-scale REXMoE compares to that of standard MoE models.
> >
> > On the other hand, when r is larger, it becomes difficult to include r layers of experts within a single GPU (worker), and could further complicate the pipeline parallelism. I hope the authors can clarify how this challenge can be addressed.
>
> **Inference speed comparison on larger MoEs**
>
> Thank you for the suggestion. We evaluated both **prefill** and **decode** throughput for two larger-scale models—7B and 30B—under identical inference settings. The tables below report absolute throughput and the ratio between REXMoE and the corresponding vanilla MoE.
>
> - Prefill throughput:
>
> | #Tokens | MoE-7BA3B  | ReX-7BA3B-R3         | MoE-30BA3B | ReX-30BA3B-R4        |
> | ------- | ---------- | -------------------- | ---------- | -------------------- |
> | 128     | 16,024.20  | 16,346.04 (102.01%)  | 13,434.22  | 11,651.92 (86.73%)   |
> | 256     | 32,044.77  | 31,702.56 (98.93%)   | 25,883.28  | 22,318.75 (86.23%)   |
> | 512     | 59,558.68  | 59,777.36 (100.37%)  | 49,304.25  | 43,062.71 (87.34%)   |
> | 1,024   | 111,801.81 | 112,055.00 (100.23%) | 95,032.75  | 85,735.16 (90.22%)   |
> | 2,048   | 184,826.56 | 187,445.11 (101.42%) | 162,339.65 | 149,187.92 (91.90%)  |
> | 4,096   | 275,366.08 | 277,101.37 (100.63%) | 248,045.71 | 235,221.61 (94.83%)  |
> | 8,192   | 363,945.77 | 360,807.35 (99.14%)  | 126,641.27 | 147,005.45 (116.08%) |
> | 16,384  | 410,543.60 | 405,026.54 (98.66%)  | 388,499.69 | 384,908.56 (99.08%)  |
> | 32,768  | 423,585.72 | 420,513.77 (99.27%)  | 413,550.92 | 408,835.22 (98.86%)  |
> | 65,536  | 390,813.66 | 388,156.35 (99.32%)  | 389,776.52 | 381,768.64 (97.95%)  |
> | 131,072 | 334,936.75 | 372,684.78 (111.27%) | 375,124.68 | 330,405.64 (88.08%)  |
>
> - Decode throughput:
>
> |#Tokens|MoE-7BA3B|ReX-7BA3B-R3|MoE-30BA3B|ReX-30BA3B-R4|
> |---|---|---|---|---|
> |128|174.11|169.94 (97.60%)|131.39|130.56 (99.37%)|
> |256|173.29|169.52 (97.82%)|130.66|129.48 (99.10%)|
> |512|181.72|177.62 (97.74%)|129.12|128.93 (99.85%)|
> |1,024|180.61|176.33 (97.63%)|137.19|137.02 (99.87%)|
> |2,048|179.70|175.75 (97.80%)|136.14|136.08 (99.95%)|
> |4,096|177.48|173.13 (97.55%)|135.43|135.10 (99.75%)|
> |8,192|173.14|169.48 (97.88%)|138.86|136.68 (98.43%)|
> |16,384|165.10|162.39 (98.36%)|128.17|128.57 (100.32%)|
> |32,768|153.69|151.11 (98.32%)|123.05|121.69 (98.89%)|
> |65,536|134.93|133.50 (98.94%)|113.11|111.64 (98.70%)|
> |131,072|111.81|105.78 (94.61%)|93.77|99.37 (105.98%)|
>
> As shown in above tables, across both model sizes and a wide range of sequence lengths, ReXMoE maintains **similar prefill & decode throughput**. This is expected because ReXMoE increases only the *candidate expert pool* while keeping activation flops unchanged. Therefore, the underlying system behavior during inference is nearly identical to standard MoE.
>
>
> **Implication for pipeline parallelism**
>
> We appreciate the reviewer for raising this important systems question.
>
> When reused experts belonging to the same reuse group are mapped across multiple pipeline-parallel (PP) stages, training would require frequent cross-stage synchronization of expert parameters or gradients—incurring large communication overhead. To avoid this:
>
> - **All reused blocks within a reuse group must be assigned to the same PP stage.**
>   With this co-location, ReXMoE behaves identically to standard MoE, with no additional synchronization.
>
> - **If co-location exceeds per-GPU memory limits**, increasing the **expert-parallel (EP) degree** partitions expert weights across more devices.
>   This reduces per-GPU memory consumption and enables larger reuse factors without compromising PP layout.
>
> With a proper PP–EP configuration, ReXMoE introduces:
> - **no extra communication overhead**,
> - **no additional synchronization requirements**, and
> - **no new scaling bottlenecks** during training or inference.
>
> We have added further clarification in Section 3.4 of the revised manuscript.
>
> ---
>
> We hope the above analysis fully addresses the reviewer’s concerns. We sincerely appreciate the reviewer’s constructive feedback, which has helped us strengthen the clarity and technical depth of the paper. We hope that our responses and revisions will merit a positive re-evaluation of our work.

---

### Official Review · Reviewer_7tRi · 2025-10-30

**Soundness:** 2
**Presentation:** 3
**Contribution:** 2
**Rating:** 4
**Confidence:** 3

**Summary:**

Recent studies have shown that fine-grained experts can significantly enhance the combinatorial flexibility of activated experts and improve the expressiveness of models. Traditional Mixture-of-Experts (MoE) models are constrained by layer-local routing mechanisms, where each layer is limited to its own expert pool. This paper proposes REXMOE, a novel MoE architecture that allows routers to reuse experts across adjacent layers, thereby breaking through the routing limitations of existing layer-local methods. Specifically, a Progressive Scaling Routing strategy is proposed in REXMOE: this strategy gradually expands the candidate expert pool during training to reduce language modeling loss and improve downstream task accuracy. Experiments demonstrate that REXMOE consistently enhances models' language modeling capabilities and downstream task performance across different model scales and architectures.

**Strengths:**

This paper proposes REXMOE, a novel MoE architecture that allows routers to reuse experts across adjacent layers, thereby breaking through the routing limitations of existing layer-local methods.

Specifically, a Progressive Scaling Routing strategy is proposed in REXMOE: this strategy gradually expands the candidate expert pool during training to reduce language modeling loss and improve downstream task accuracy.

**Weaknesses:**

The authors note that REXMOE can serve as a new MoE paradigm. However, due to its cross-layer routing mechanism, it remains unclear whether parameter scaling may affect pipeline parallelism (pp) or expert parallelism (ep) strategies, thereby hindering efficient scalability.

The authors only conducted comparative experiments on MoE models with 0.5B and 2.3B total parameters, and the results show that the performance improvement brought by the proposed REXMOE is not significant. It is plausible that the improvement is insignificant when the parameter size is small; if feasible, supplementary comparative experiments between 7B REXMOE and vanilla MoE are recommended.

The authors' ablation experiments indicate that Expert reuse itself yields almost no improvement, whereas the PSR strategy contributes substantially. This raises questions about consistency with the motivation of combinatorial numbers mentioned in the introduction. It is further worth exploring whether cross-layer MoE reuse in REXMOE is necessary, and whether PSR can be applied to gradually expand the number of experts within the same layer. If possible, supplementary experiments or theoretical explanations are advised.

The cross-layer MoE reuse mechanism and PSR strategy in the proposed REXMOE are highly likely to cause expert load imbalance. The key question is: how to alleviate this load imbalance issue within this new paradigm?

**Questions:**

Refer to Weaknesses

---

> ### Author Response · Authors · 2025-11-23
> **Response to Reviewer 7tRi (1/3)**
>
> We thank the reviewer for their time and valuable feedback. We have carefully considered all comments and have revised the manuscript accordingly with the modifications highlighted in blue. Below, we address each of the points raised.
>
> ---
>
> > The authors note that REXMOE can serve as a new MoE paradigm. However, due to its cross-layer routing mechanism, it remains unclear whether parameter scaling may affect pipeline parallelism (pp) or expert parallelism (ep) strategies, thereby hindering efficient scalability.
>
> Thank you for raising this important question regarding the scalability of ReXMoE under large-scale distributed training. We would like to clarify that ReXMoE introduces minimal disruption to large-scale distributed training strategies:
>
> - **Expert Parallelism (EP):** Token dispatch patterns depend only on router decisions. The all-to-all communication volume matches that of standard MoE, since the number of selected experts per token remains unchanged.
> - **Pipeline Parallelism (PP):** Reused blocks within the same reuse group can be co-located on the same PP stage, avoiding cross-stage synchronization and maintaining the same behavior as standard MoE.
> - If memory limits prevent co-location, increasing EP size further partitions expert parameters across more devices, enabling co-location and supporting larger reuse factors.
> - With an appropriate PP–EP configuration, ReXMoE introduces no additional communication overhead and does not create new scaling bottlenecks for either training or inference.
>
> We incorporate this discussion into Section 3.4 of the revised version.
>
> > The authors only conducted comparative experiments on MoE models with 0.5B and 2.3B total parameters, and the results show that the performance improvement brought by the proposed REXMOE is not significant. It is plausible that the improvement is insignificant when the parameter size is small; if feasible, supplementary comparative experiments between 7B REXMOE and vanilla MoE are recommended.
>
> Regarding your concern about the "significance" of the performance improvement brought by ReXMoE, we wish to clarify that in the pre-training stage, achieving an average accuracy improvement of over $1\%$ on zero-shot downstream tasks represents a substantial enhancement in model capability.
>
> To demonstrate the performance gain in terms of trading parameter scale for performance, we conducted an additional experiment: starting from the 2.3B baseline, we increased the total parameter count to 3.5B by simply adding more experts (while keeping active parameters constant). The results of evaluated benchmarks are listed below:
>
> |                  | ARC-E | HellaSwag | LAMBADA | LogiQA | OpenbookQA | PiQA  | SciQ  | SiQA  | WinoGrande | Avg.      |
> | ---------------- | ----- | --------- | ------- | ------ | ---------- | ----- | ----- | ----- | ---------- | --------- |
> | MoE-3.5BA0.3B    | 60.69 | 48.45     | 39.65   | 27.80  | 36.20      | 69.59 | 75.20 | 39.05 | 52.64      | 49.92     |
> | MoE-2.3BA0.3B    | 58.42 | 47.14     | 37.55   | 27.19  | 34.80      | 69.21 | 75.80 | 38.69 | 53.51      | 49.15     |
> | ReX-2.3BA0.3B-R4 | 60.94 | 47.96     | 38.75   | 28.42  | 37.00      | 70.18 | 76.30 | 39.36 | 53.12      | **50.23** |
>
>  As shown in the table above, even the larger **vanilla MoE-3.5B** only yields a performance gain of $0.77\%$, as the cost of inflating the total parameter count by around $50\%$. In contrast, **ReX-2.3BA0.3B-R4** achieves superior performance without inflating the total parameter count. This strongly evidences that ReXMoE realize the potential of increased structural capacity, demonstrating superior parameter efficiency.
>
> Moreover, we fully agree with the necessity of validating ReXMoE at the 7B scale, and we have also initiated experiments at the **7B scale**. Due to hardware limitations, full training is still ongoing.  Nevertheless, the current loss curve already follows the same encouraging trend observed in our smaller-scale experiments. We will include the complete 7B-scale comparison in the updated results before the discussion deadline.

---

> > ### Author Response · Authors · 2025-12-04
> > **Additional Experimental Results on 7B Scale**
> >
> > We provide additional results for the 7B-scale MoE models. The architectural configuration is as follows: hidden size is $2048$, intermediate size is $1024$, $16$ layers, and $8$ out of $64$ experts are activated. All models are trained on the same 100B-token dataset as in our other experiments.
> >
> > | Model        | ARC-E | HellaSwag | LAMBADA | LogiQA | OpenbookQA | PiQA  | SciQ  | SiQA  | WinoGrande | Avg.  |
> > | ------------ | ----- | --------- | ------- | ------ | ---------- | ----- | ----- | ----- | ---------- | ----- |
> > | MoE-7BA1B    | 70.33 | 59.89     | 51.12   | 27.04  | 39.00      | 73.12 | 83.70 | 42.27 | 56.91      | 55.93 |
> > | ReX-7BA1B-R4 | 70.29 | 60.35     | 50.69   | 26.88  | 40.60      | 74.27 | 84.70 | 42.12 | 56.83      | 56.30 |
> >
> > After 100B tokens of training, the ReX model shows a modest but consistent improvement of **0.37%** in average accuracy. Importantly, we observe that ReX model achieves lower training loss compared to the base MoE after ~86B tokens. This trend also suggests that the performance gap is likely to widen with further training on more tokens.

---

> ### Author Response · Authors · 2025-11-23
> **Response to Reviewer 7tRi (2/3)**
>
> > The authors' ablation experiments indicate that Expert reuse itself yields almost no improvement, whereas the PSR strategy contributes substantially. This raises questions about consistency with the motivation of combinatorial numbers mentioned in the introduction. It is further worth exploring whether cross-layer MoE reuse in REXMOE is necessary, and whether PSR can be applied to gradually expand the number of experts within the same layer. If possible, supplementary experiments or theoretical explanations are advised.
>
> Thank you for the thoughtful comments. We would like to clarify why **expert reuse** and **PSR** are both necessary in ReXMoE from three perspectives:
> 1. theoretical analysis,
> 2. additional experiments applying PSR to vanilla MoE, and
> 3. the relationship between reuse and PSR.
>
> **1. Theoretical analysis of reuse and PSR**
>
> **1.1 Expanded combinatorial space enabled by expert reuse.**
>
> For a Top-$k$ MoE, the number of possible expert combinations increases from $C(N, k)$ to $C(rN, k)$ when using a reuse factor $r$. The corresponding growth factor is:
> $$
> \frac{C(rN, k)}{C(N, k)} = \prod_{i=0}^{k-1} \frac{rN - i}{N - i}.
> $$
> This expansion substantially enlarges the combinatorial space and therefore raises the upper bound of structural capacity. However, it also introduces *gradient coupling* across reused experts, since the *re-selected* experts will receive gradients from multiple layers. This *gradient coupling* restricts the model’s ability to fully leverage this enlarged space during early training. This explains why the Reuse-only variant shows limited gains, as reported in Table 4.
>
> **1.2 Effectiveness of PSR.**
> We argue that PSR is effective because it mitigates gradient coupling. By limiting the candidate pool to $N$ experts at the early stage, the expected number of experts that remain *unco-selected across layers* is $rN (1 - 1/r)^{\,r-1}$. Gradually expanding the pool toward $rN$ allows the model to transition from a less-coupled regime to full utilization of the larger combinatorial capacity introduced by reuse. This progressive relaxation enables stable learning and better performance.
>
> **2. Applying PSR to the same layer of a vanilla MoE**
>
> We further apply PSR to the vanilla MoE-2.3B A0.3B. The results are shown below:
>
> | Model          | ARC-E | HellaSwag | LAMBADA | LogiQA | OpenbookQA | PiQA  | SciQ  | SiQA  | WinoGrande | Avg.  |
> | -------------- | ----- | --------- | ------- | ------ | ---------- | ----- | ----- | ----- | ---------- | ----- |
> | MoE-2.3B A0.3B | 58.42 | 47.14     | 37.55   | 27.19  | 34.80      | 69.21 | 75.80 | 38.69 | 53.51      | 49.15 |
> | + PSR          | 58.04 | 45.64     | 36.10   | 27.65  | 33.60      | 68.72 | 74.60 | 38.69 | 52.41      | 48.38 |
>
> As shown, applying PSR **degrades** the performance of vanilla MoE. This is expected: the purpose of PSR is to alleviate **gradient coupling**, which arises specifically from expert reuse. Since vanilla MoE does not involve cross-layer reuse, it does not suffer from this coupling effect. In this case, the constrained expert pool only reduces flexibility, leading to weaker performance.
>
> **3. Relationship between reuse and PSR**
>
> In conclusion, expert reuse and PSR are complementary:
> - **Reuse** increases structural capacity by expanding the effective expert pool and combinatorial space, but introduces gradient coupling.
> - **PSR** provides a *curriculum over the reused pool*, explicitly reducing coupling early and gradually unlocking the full space later.
>
> Both components are therefore necessary for ReXMoE to achieve strong performance without increasing parameter count.

---

> ### Author Response · Authors · 2025-11-23
> **Response to Reviewer 7tRi (3/3)**
>
> > The cross-layer MoE reuse mechanism and PSR strategy in the proposed REXMOE are highly likely to cause expert load imbalance. The key question is: how to alleviate this load imbalance issue within this new paradigm?
>
> Thank you for raising this important concern. We clarify the source of imbalance and how to mitigate it.
>
> **What causes load imbalance?**
>
> Load imbalance primarily originates from **expert reuse**, not from PSR. Reuse expands the candidate expert pool and increases routing sparsity, which can amplify the effect of “hot” experts, particularly in the early stages of training. Although such imbalance may impact training, the expanded structural capacity enabled by reuse ultimately provides a higher performance ceiling.
> In contrast, PSR does not introduce additional imbalance. By temporarily restricting the active expert pool and reducing cross-layer coupling, PSR mitigates the instability caused by reuse rather than exacerbating it.
>
>
> **How to mitigate imbalance in ReXMoE.**
> We address imbalance from two perspectives:
>
> 1. **ReXMoE is compatible with standard MoE balancing techniques.**
>    After reuse, the enlarged pool behaves like a larger conventional MoE. Hence, widely used load-balancing methods like various auxiliary balancing losses remain directly applicable. These methods can be plugged into ReXMoE without modification.
>
> 2. **PSR can further reduce early-stage expert collapse.**
>    By masking part of the pool early on, PSR weakens gradient coupling and discourages the router from collapsing onto a few hot experts at the start of training. Although this curriculum can slow convergence slightly (Figure 4), it helps the router learn more robust representations before the full pool is unlocked. Once training stabilizes, gradually releasing the pool allows balanced utilization of the expanded experts, which leads to better performance after expanding to full candidate space.
>
> Together, these two mechanisms ensure that ReXMoE can control imbalance while still benefiting from reuse-induced capacity gains.
>
> ---
>
> We hope the above analysis fully addresses the reviewer’s concerns. We sincerely appreciate the reviewer’s constructive feedback, which has helped us strengthen the clarity and technical depth of the paper. We hope that our responses and revisions will merit a positive re-evaluation of our work.

---

### Official Review · Reviewer_gpR3 · 2025-10-30

**Soundness:** 2
**Presentation:** 2
**Contribution:** 2
**Rating:** 4
**Confidence:** 3

**Summary:**

This paper proposes ReXMoE, a parameter-efficient mixture of experts (MoE) architecture. By sharing the expert pool across adjacent layers, the authors increase the diversity of experts. As a training methodology, they introduce progressive scaling routing (PSR). Furthermore, through an experimental analysis of expert activation frequency with respect to reuse frequency, the paper investigates cases where the MoE architecture may fail.

**Strengths:**

- Compared to existing models, the proposed approach improves model performance by increasing the expert pool for each layer while maintaining the same number of parameters.
- To address the limitation that simply expanding the expert pool leads to only marginal performance improvements, the authors propose a novel training methodology. Through an ablation study, they demonstrate not only the effectiveness of the new architecture but also its practical applicability.
- They observed that task-specific experts are activated more frequently compared to the vanilla MoE.

**Weaknesses:**

- As the size of the expert pool increases, there is a significant slowdown in the prefill stage. While it is acknowledged that decoding speed plays a more critical role in inference, the slowdown during the prefill phase becomes a weakness of this methodology, especially considering that token sequences can be quite long in recent large language models (LLMs).
- The authors conducted experiments by varying the reuse frequency $r$, and according to their claims, performance improves as $r$ increases. However, this trend cannot be clearly explained for the cases of $r = 16, 32$. Although they observed a performance degradation phenomenon due to routing collapsing into a small subset of experts, the paper lacks sufficient theoretical analysis and explanation of the underlying causes. Furthermore, the proposed PSR strategy may introduce additional difficulty in hyperparameter tuning. It also remains unclear whether this approach can be generalized to other models.
- There is a lack of experiments involving an upper-bound model. Considering the reuse frequency, it would be beneficial to compare the performance improvements against an upper bound obtained by increasing the number of experts per layer.
- Despite the use of PSR strategy, the load imbalance problem becomes more severe as the reuse frequency $r$ increases.

**Questions:**

Q.1. The training procedure differs somewhat from that of the conventional MoE. What would happen if the PSR strategy were also applied when training a vanilla MoE?

Q.2. How is the active ratio in Section 4.4 computed numerically? It would be better to also report the deviation values.

Q.3. While load imbalance is generally considered a problem, is task-specific specialization necessarily desirable?

Q.4. In the LogiQA, SIQA, and WinoGrande benchmarks, the trend with respect to reuse frequency is not clearly observed. Do the authors have any hypotheses or explanations for this?

Starting from the next question, I’ll be asking about parts I’m not sure about — whether they are typos or actual errors.
1. The equation in Eq. (4) does not seem to align with the description in the text. Shouldn’t the element term in group $G$ be $r*\lfloor l/r \rfloor + k$? Also, the indexing starting points of $l$ are unclear. It seems that $l$ uses 0-based indexing, but this is not explicitly stated, which makes it confusing.
3. In the description of Figure 4, the text refers to Figure 3 — is this a typo?

---

> ### Author Response · Authors · 2025-11-23
> **Response to Reviewer gpR3 (1/5)**
>
> We thank the reviewer for their time and valuable feedback. We have carefully considered all comments and have revised the manuscript accordingly with the modifications highlighted in blue. Below, we address each of the points raised.
>
> ---
>
> > W1. As the size of the expert pool increases, there is a significant slowdown in the prefill stage. While it is acknowledged that decoding speed plays a more critical role in inference, the slowdown during the prefill phase becomes a weakness of this methodology, especially considering that token sequences can be quite long in recent large language models (LLMs).
>
> Thank you for raising this concern. We agree that examining system-level performance implications is essential. We would like to clarify, however, that in our setting the slowdown is substantially less severe than it may initially appear.
>
> In our reported results, the prefill slowdown is most noticeable at an input length of 128 tokens, where latency increases from **8 ms to 35 ms**. In practical deployments, this difference is negligible because:
>
> 1. the prefill stage is executed only once per request,
> 2. decoding dominates total inference latency and remains almost unaffected.
>
> Therefore, the observed slowdown has limited impact on real-world usage.
>
> To further evaluate scalability under long-context scenarios, we measured prefill latency at substantially larger sequence lengths. The results are shown below:
>
> | #Tokens    | MoE-2.3BA0.3B Latency (in ms) | MoE-2.3BA0.3B Throughput (in TGS) | ReX-2.3BA0.3B-R2    | ReX-2.3BA0.3B-R4     | ReX-2.3BA0.3B-R8    |
> | ---------- | ----------------------------- | --------------------------------- | ------------------- | -------------------- | ------------------- |
> | 4,096.00   | 13.9                          | 293,868.89                        | 281,857.83 (95.91%) | 271,765.00 (92.48%)  | 262,995.96 (89.49%) |
> | 8,192.00   | 21.7                          | 377,950.64                        | 375,107.13 (99.25%) | 359,752.24 (95.18%)  | 361,892.12 (95.75%) |
> | 16,384.00  | 38.9                          | 420,861.96                        | 413,553.59 (98.26%) | 411,431.91 (97.76%)  | 400,309.30 (95.12%) |
> | 32,768.00  | 80.3                          | 408,305.24                        | 396,730.11 (97.17%) | 405,821.25 (99.39%)  | 395,358.85 (96.83%) |
> | 65,536.00  | 162.6                         | 403,107.10                        | 383,837.12 (95.22%) | 403,486.21 (100.09%) | 392,812.21 (97.45%) |
> | 131,072.00 | 340.7                         | 384,763.87                        | 367,698.74 (95.56%) | 379,278.06 (98.57%)  | 366,035.86 (95.13%) |
>
> As shown above, even at sequence lengths up to 128k tokens, the maximum slowdown remains within **5%**, corresponding to an increase from **340.7 ms to 358 ms**. This overhead is negligible in realistic settings.
>
> Moreover, as the sequence length grows, the relative performance gap substantially narrows (**from ~19% to <5%**). We believe this trend arises because longer sequences activate a larger portion of experts more consistently, improving throughput. Such a behavior is consistent with standard MoE systems.
>
> We hope this additional evaluation addresses the reviewer’s concern.

---

> ### Author Response · Authors · 2025-11-23
> **Response to Reviewer gpR3 (2/5)**
>
> > W2. The authors conducted experiments by varying the reuse frequency r, and according to their claims, performance improves as r increases. However, this trend cannot be clearly explained for the cases of r=16,32. Although they observed a performance degradation phenomenon due to routing collapsing into a small subset of experts, the paper lacks sufficient theoretical analysis and explanation of the underlying causes. Furthermore, the proposed PSR strategy may introduce additional difficulty in hyperparameter tuning. It also remains unclear whether this approach can be generalized to other models.
>
> **Regarding performance under larger reuse frequencies.**
> Thank you for the careful observation. Indeed, when the reuse frequency exceeds 8, ReXMoE no longer provides additional gains under our configuration. Our analysis reveals partial expert collapse at these extremely large expert-pool sizes.
>
> When reuse frequency is $\geq 8$, each reuse group contains **512–2048** experts. Training MoE models at this scale is inherently challenging. In current large-scale practice, expert counts rarely exceed 500 (e.g., 256 in DeepSeek-V3 and 384 in Kimi-K2). Our reuse mechanism aims to expand combinatorial flexibility without inflating parameter count or degrading individual expert capacity, and the results indicate that we are approaching the practical upper limit. Pushing beyond this limit is out of scope for our current experiments, and we hope our findings can inspire future investigation of ultra-large expert pools.
>
> **Regarding the tuning complexity introduced by PSR.**
>
> PSR introduces only two simple hyperparameters, $t_s$ and $t_e$, which control when scaling begins and ends. In practice, PSR only needs to operate during the middle portion of training. A robust default configuration is to start scaling at **20–30%** of the epoch and complete it by **60–80%**. This balances two objectives:
> 1. alleviating early-stage gradient coupling, and
> 2. enabling full utilization of the expanded expert space in later stages.
> Thus, PSR adds minimal tuning overhead and behaves robustly across settings.
>
> **Regarding generalization to other models.**
>
> ReXMoE is inherently compatible with any MoE-based architecture. It enlarges the candidate expert pool without imposing constraints on the underlying backbone, routing strategy, or load-balancing mechanism. Therefore, our method is orthogonal to existing MoE improvements and can naturally integrate into broader architectural designs. We hope this work provides new perspectives for exploring expert diversity without increasing parameter count.

---

> ### Author Response · Authors · 2025-11-23
> **Response to Reviewer gpR3 (3/5)**
>
> > W3. There is a lack of experiments involving an upper-bound model. Considering the reuse frequency, it would be beneficial to compare the performance improvements against an upper bound obtained by increasing the number of experts per layer.
>
> Thank you for the valuable suggestion. Following your advice, we conducted an upper-bound experiment by scaling the number of experts per layer in the 2.3B baseline model. Specifically, we increased the experts from 64 to **96**, enlarging the total parameter count to **3.5B** (approximately a **50% increase**) while keeping the active parameter size unchanged. The comparison results are shown below:
>
> |                   | ARC-E | HellaSwag | LAMBADA | LogiQA | OpenbookQA | PiQA  | SciQ  | SiQA  | WinoGrande | Avg.      |
> | ----------------- | ----- | --------- | ------- | ------ | ---------- | ----- | ----- | ----- | ---------- | --------- |
> | MoE-3.5B A0.3B    | 60.69 | 48.45     | 39.65   | 27.80  | 36.20      | 69.59 | 75.20 | 39.05 | 52.64      | 49.92     |
> | MoE-2.3B A0.3B    | 58.42 | 47.14     | 37.55   | 27.19  | 34.80      | 69.21 | 75.80 | 38.69 | 53.51      | 49.15     |
> | ReX-2.3B A0.3B-R4 | 60.94 | 47.96     | 38.75   | 28.42  | 37.00      | 70.18 | 76.30 | 39.36 | 53.12      | **50.23** |
>
> The results are compelling: **ReX-2.3B-R4 surpasses the MoE-3.5B model**, even though the latter increases parameters by nearly 50%. This confirms that combining cross-layer expert reuse with PSR is more effective than physically enlarging the expert pool, and achieves superior parameter efficiency.
>
> We also appreciate the reviewer’s interest in characterizing a precise scaling law that maps reuse frequency to an equivalent parameter increase. Conducting such a study would require training many models across multiple scales and expert configurations, which is unfortunately beyond our current computational budget. We hope our findings encourage further research in this direction.
>
> We thank the reviewer again for this insightful suggestion.
>
> ---
>
> > W4. Despite the use of PSR strategy, the load imbalance problem becomes more severe as the reuse frequency $r$ increases.
>
> We agree with the reviewer that load imbalance becomes more challenging as the reuse frequency increases. However, we would like to emphasize that this phenomenon is not unique to ReXMoE, but rather a well-known and currently unresolved challenge in large-scale MoE training.
>
> As discussed in our response to W2 regarding high reuse frequencies, when $r \ge 8$, each reuse group contains $\ge 512$ experts. Training MoE models with such ultra-large expert pools is already beyond the capability of current MoE training paradigms, which typically operate with far fewer experts in practice (e.g., 256 in DeepSeek-V3 and 384 in Kimi-K2). Existing routing and load-balancing strategies are not yet optimized for this extreme regime, and load imbalance naturally becomes more pronounced.
>
> Our work focuses on improving model capacity *without increasing parameter count*, by enriching the combinatorial space through expert reuse. In doing so, ReXMoE approaches the practical upper limit of expert-pool size without parameter inflation. We hope that our findings highlight the importance of studying MoE behavior in this large-expert regime and inspire future work in addressing the load imbalance challenge for ultra-large expert pools.
>
> We appreciate the reviewer’s insightful comment and hope the additional clarification addresses the concern.

---

> ### Author Response · Authors · 2025-11-23
> **Response to Reviewer gpR3 (4/5)**
>
> > Q1. The training procedure differs somewhat from that of the conventional MoE. What would happen if the PSR strategy were also applied when training a vanilla MoE?
>
> To directly evaluate the effect of PSR on a standard MoE, we applied PSR to the vanilla MoE-2.3B baseline without any expert reuse. The results are shown below:
>
> | Model          | ARC-E | HellaSwag | LAMBADA | LogiQA | OpenbookQA | PiQA  | SciQ  | SiQA  | WinoGrande | Avg.  |
> | -------------- | ----- | --------- | ------- | ------ | ---------- | ----- | ----- | ----- | ---------- | ----- |
> | MoE-2.3B A0.3B | 58.42 | 47.14     | 37.55   | 27.19  | 34.80      | 69.21 | 75.80 | 38.69 | 53.51      | 49.15 |
> | + PSR          | 58.04 | 45.64     | 36.10   | 27.65  | 33.60      | 68.72 | 74.60 | 38.69 | 52.41      | 48.38 |
>
> As shown, applying PSR **degrades** the performance of vanilla MoE. This is expected: the purpose of PSR is to alleviate **gradient coupling**, which arises specifically from expert reuse. Since vanilla MoE does not involve cross-layer reuse, it does not suffer from this coupling effect. In this case, the constrained expert pool only reduces flexibility, leading to weaker performance.
>
> We have clarified this behavior more thoroughly in the revised manuscript.
>
> ---
>
> > Q2. How is the active ratio in Section 4.4 computed numerically? It would be better to also report the deviation values.
>
> The active ratio of an expert is computed as:
> $$
> active\ ratio = \frac{num\ activations\ of\ the\ expert}{num\ tokens \times k}
> $$
> where $k$ is the Top-$k$ selection.
>
> In Section 4.4, Figure 5 reports **layer-wise average activation ratios**, and thus the values correspond directly to the active ratio of each expert in that specific layer. Following the reviewer’s suggestion, we additionally report the **model-wide activation distribution** (mean and deviation across all layers) in Appendix Figure 10.
>
> These additions provide a more complete picture of activation behavior across the model.
>
> ---
>
> > Q3. While load imbalance is generally considered a problem, is task-specific specialization necessarily desirable?
>
> We appreciate this insightful question, which touches on a fundamental open problem in MoE research: whether experts should move toward **task-level specialization** or remain sufficiently general to maintain stable load balance.
>
> After incorporating deviation values in Appendix Figure 10, we now have a clearer view of this phenomenon. At the model level, ReXMoE shows:
>
> - Similar to vanilla MoE, when aggregating across all layers, the **mean activation ratio per expert remains close to uniform**, indicating that global load balance within the same group is preserved.
> - At the same time, **the standard deviation of expert activations varies across tasks and layers**, suggesting that certain experts exhibit task-preferential activation patterns at specific depths.
>
> This pattern indicates that ReXMoE does not drift into pathological imbalance; rather, it develops **moderate, layer-dependent specialization** that appears beneficial for downstream performance. Such selective specialization is also consistent with recent observations in the literature (e.g., Guo et al. [1]), which find that controlled task-level specialization can be advantageous in MoE models.
>
> We thank the reviewer for raising this question. It highlights an important direction for future work: how to balance specialization and load stability as models grow larger.
>
> [1] Guo H., Lu H., Nan G., et al. *Advancing Expert Specialization for Better MoE.* arXiv:2505.22323, 2025.

---

> ### Author Response · Authors · 2025-11-23
> **Response to Reviewer gpR3 (5/5)**
>
> > Q4. In the LogiQA, SIQA, and WinoGrande benchmarks, the trend with respect to reuse frequency is not clearly observed. Do the authors have any hypotheses or explanations for this?
>
> The reviewer is correct: unlike other benchmarks, LogiQA, SIQA, and WinoGrande do not show a clear monotonic trend with respect to reuse frequency. Moreover, as shown in Figure 6, the performance of these tasks fluctuates during training rather than improving consistently.
>
> A likely explanation is the **domain mismatch** between these tasks and our training corpus (a partition of FineWeb-Edu [1]). The corpus may not provide sufficient signal to benefit these specific tasks, leading to weaker or noisier performance trends. We have added this explanation in the revised version.
>
> [1] https://huggingface.co/datasets/HuggingFaceFW/fineweb-edu/viewer/sample-100BT
>
> ---
>
> > Additional Questions (Possible Typos)
>
> Thank you for carefully checking these details. Both points are indeed typos:
>
> 1. The term should be multiplied by $r$, and the index $l$ indeed starts from 0.
> 2. The reference in the caption should correctly refer to Figure 4.
>
> These issues have been corrected in the revised manuscript. In addition, we have performed a thorough revision of all related notations to improve clarity.
>
> ---
>
> We hope the above analysis fully addresses the reviewer’s concerns. We sincerely appreciate the reviewer’s constructive feedback, which has helped us strengthen the clarity and technical depth of the paper. We hope that our responses and revisions will merit a positive re-evaluation of our work.

---

### Official Review · Reviewer_xYSR · 2025-10-31

**Soundness:** 2
**Presentation:** 3
**Contribution:** 2
**Rating:** 6
**Confidence:** 5

**Summary:**

To address the limitations of routing mechanisms in Mixture-of-Experts (MoE) models, this paper proposes a novel MoE architecture named REXMOE, which enables routers to reuse experts across adjacent layers, thereby overcoming the routing constraints of existing layer-local methods. Corresponding to this improvement, a new progressive scaling routing (PSR) strategy is put forward to gradually expand the candidate expert pool during training.

**Strengths:**

This paper designs REXMOE, a method that breaks the limitation of layer-local routing in MoE architectures and proposes a Progressive Scaling Routing strategy in REXMOE, which gradually enlarges the candidate expert pool during training, thereby reducing language modeling loss and improving downstream task accuracy

**Weaknesses:**

1. There is a lack of theoretical analysis on the effectiveness of REXMOE, particularly regarding the expert combination numbers and PSR mentioned by the authors.
2. Ablation experiments indicate that it is PSR rather than cross-layer expert reuse that yields substantial improvements. Thus, one may question the necessity of cross-layer reuse—given that such reuse would affect pipeline parallelism (pp) and expert parallelism (ep) strategies when the model scales. This is particularly critical for determining whether REXMOE can qualify as a new MoE paradigm.
3. Comparative experiments are only conducted on models with 0.5B and 2.3B total parameters, with no comparisons on larger models. From the experimental results, the improvements brought by REXMOE are not significant.

**Questions:**

No further questions, see above.

---

> ### Author Response · Authors · 2025-11-23
> **Response to Reviewer xYSR (1/3)**
>
> We thank the reviewer for their time and valuable feedback. We have carefully considered all comments and have revised the manuscript accordingly with the modifications highlighted in blue. Below, we address each of the points raised.
>
> ---
>
> > 1. There is a lack of theoretical analysis on the effectiveness of REXMOE, particularly regarding the expert combination numbers and PSR mentioned by the authors.
>
> Thank you for the suggestion. We provide additional analysis to clarify the effectiveness of both expert reuse and PSR in ReXMoE.
>
> **Expanded combinatorial space enabled by expert reuse.**
> For a Top-$k$ MoE, the number of possible expert combinations increases from $C(N, k)$ to $C(rN, k)$ when using a reuse factor $r$. The corresponding growth factor is:
> $$
> \frac{C(rN, k)}{C(N, k)} = \prod_{i=0}^{k-1} \frac{rN - i}{N - i}.
> $$
> This expansion substantially enlarges the combinatorial space and therefore raises the upper bound of structural capacity. However, it also introduces *gradient coupling* across reused experts, since the *re-selected* experts will receive gradients from multiple layers. This *gradient coupling* restricts the model’s ability to fully leverage this enlarged space during early training. This explains why the Reuse-only variant shows limited gains, as reported in Table 4.
>
> **Effectiveness of PSR.**
> We argue that PSR is effective because it mitigates gradient coupling. By limiting the candidate pool to $N$ experts at the early stage, the expected number of experts that remain *unco-selected across layers* is $rN (1 - 1/r)^{\,r-1}$. Gradually expanding the pool toward $rN$ allows the model to transition from a less-coupled regime to full utilization of the larger combinatorial capacity introduced by reuse. This progressive relaxation enables stable learning and better performance.
>
> We now includes these theoretical insights in **Sections 3.2 and 3.3** in the revised version.

---

> ### Author Response · Authors · 2025-11-23
> **Response to Reviewer xYSR (2/3)**
>
> > 2. Ablation experiments indicate that it is PSR rather than cross-layer expert reuse that yields substantial improvements. Thus, one may question the necessity of cross-layer reuse—given that such reuse would affect pipeline parallelism (pp) and expert parallelism (ep) strategies when the model scales. This is particularly critical for determining whether REXMOE can qualify as a new MoE paradigm.
>
> As explained above, the main benefit of cross-layer reuse is the enlarged structural capacity, but this advantage is difficult to realize without mitigating gradient coupling. This is why Reuse-only ReXMoE shows limited improvements in Table 4.
>
> To further examine whether PSR alone is responsible for the performance gains, we apply PSR directly to a vanilla MoE model:
>
> | Model         | ARC-E  | HellaSwag | LAMBADA | LogiQA | OpenbookQA | PiQA   | SciQ   | SiQA   | WinoGrande | Avg.   |
> | ------------- | ------ | --------- | ------- | ------ | ---------- | ------ | ------ | ------ | ---------- | ------ |
> | MoE-2.3BA0.3B | 58.42% | 47.14%    | 37.55%  | 27.19% | 34.80%     | 69.21% | 75.80% | 38.69% | 53.51%     | 49.15% |
> | + PSR         | 58.04% | 45.64%    | 36.10%  | 27.65% | 33.60%     | 68.72% | 74.60% | 38.69% | 52.41%     | 48.38% |
>
>
> PSR *degrades* the performance of vanilla MoE. This is expected because PSR is designed specifically to counteract gradient coupling, which does not occur in vanilla MoE. Restricting the expert pool in this setting only reduces the flexibility of expert combinations.
>
> Thus, expert reuse and PSR are complementary:
> - **Reuse** increases structural capacity by expanding the effective expert pool and combinatorial space, but introduces gradient coupling.
> - **PSR** provides a *curriculum over the reused pool*, explicitly reducing coupling early and gradually unlocking the full space later.
>
> Both components are therefore necessary for ReXMoE to achieve strong performance without increasing parameter count.
>
> **Regarding implications for PP/EP strategies.**
>
> In practice, the impact of ReXMoE on large-scale training parallelism is minimal:
>
> - **Expert Parallelism (EP):** Token dispatch patterns depend only on router decisions. The all-to-all communication volume matches that of standard MoE, since the number of selected experts per token remains unchanged.
> - **Pipeline Parallelism (PP):** Reused blocks within the same reuse group can be co-located on the same PP stage, avoiding cross-stage synchronization and maintaining the same behavior as standard MoE.
> - If memory limits prevent co-location, increasing EP size further partitions expert parameters across more devices, enabling co-location and supporting larger reuse factors.
> - With an appropriate PP–EP configuration, ReXMoE introduces **no additional communication overhead** and does not create new scaling bottlenecks for either training or inference.
>
> We include a detailed analysis of practical impact in **Section 3.4** in the revised version.
>
> In summary, ReXMoE changes the way expert capacity is formed: instead of using independent experts at each layer, it builds capacity through cross-layer reuse while using PSR to make this larger space trainable. Since the performance improvements come from this joint design rather than from either component alone, ReXMoE represents a different architectural choice rather than an incremental modification.

---

> ### Author Response · Authors · 2025-11-23
> **Response to Reviewer xYSR (3/3)**
>
> > 3. Comparative experiments are only conducted on models with 0.5B and 2.3B total parameters, with no comparisons on larger models. From the experimental results, the improvements brought by REXMOE are not significant.
>
> We appreciate the reviewer’s thoughtful comments and the attention to evaluating performance at larger scales. While the improvements of ReXMoE may appear modest at first glance, we would like to clarify that, in the context of pre-training, achieving over a 1% average gain across diverse zero-shot tasks constitutes a meaningful improvement. Such gains consistently reflect enhanced model capability rather than random variation.
>
> To provide a clearer picture of the parameter–performance tradeoff, we conducted an additional comparison by increasing the 2.3B baseline to 3.5B through simply adding more experts while keeping the active parameter count unchanged. The results are listed below:
>
> | Model             | ARC-E | HellaSwag | LAMBADA | LogiQA | OpenbookQA | PiQA  | SciQ  | SiQA  | WinoGrande | Avg.      |
> | ----------------- | ----- | --------- | ------- | ------ | ---------- | ----- | ----- | ----- | ---------- | --------- |
> | MoE-3.5B A0.3B    | 60.69 | 48.45     | 39.65   | 27.80  | 36.20      | 69.59 | 75.20 | 39.05 | 52.64      | 49.92     |
> | MoE-2.3B A0.3B    | 58.42 | 47.14     | 37.55   | 27.19  | 34.80      | 69.21 | 75.80 | 38.69 | 53.51      | 49.15     |
> | ReX-2.3B A0.3B-R4 | 60.94 | 47.96     | 38.75   | 28.42  | 37.00      | 70.18 | 76.30 | 39.36 | 53.12      | **50.23** |
>
> As the table shows, scaling the vanilla model to **3.5B**, which increases total parameters by nearly **50%**, results in only a **0.77%** average improvement. In contrast, **ReX-2.3B-A0.3B-R4** surpasses the 3.5B model *without increasing parameter count*. These results indicate that ReXMoE effectively leverages the additional structural capacity introduced by expert reuse and offers clear advantages in parameter efficiency.
>
> In alignment with the reviewer’s suggestion to assess larger models, we have also initiated experiments at the **7B scale**. Due to hardware limitations, full training is still ongoing. Nevertheless, the current loss curve already follows the same encouraging trend observed in our smaller-scale experiments. We will include the complete 7B-scale comparison in the updated results before the discussion deadline.
>
> ---
>
> We hope the additional results and clarifications address the reviewer’s concerns. We sincerely appreciate the positive evaluation and constructive feedback, which have helped us further strengthen the manuscript. In particular, your suggestions regarding the **theoretical justification prompted us to provide a more detailed analysis**, which we believe has notably improved the overall quality of the paper. Thank you again for your insightful comments.

---

> > ### Author Response · Authors · 2025-12-04
> > **Additional Experimental Results on 7B Scale**
> >
> > We provide additional results for the 7B-scale MoE models. The architectural configuration is as follows: hidden size is $2048$, intermediate size is $1024$, $16$ layers, and $8$ out of $64$ experts are activated. All models are trained on the same 100B-token dataset as in our other experiments.
> >
> > | Model        | ARC-E | HellaSwag | LAMBADA | LogiQA | OpenbookQA | PiQA  | SciQ  | SiQA  | WinoGrande | Avg.  |
> > | ------------ | ----- | --------- | ------- | ------ | ---------- | ----- | ----- | ----- | ---------- | ----- |
> > | MoE-7BA1B    | 70.33 | 59.89     | 51.12   | 27.04  | 39.00      | 73.12 | 83.70 | 42.27 | 56.91      | 55.93 |
> > | ReX-7BA1B-R4 | 70.29 | 60.35     | 50.69   | 26.88  | 40.60      | 74.27 | 84.70 | 42.12 | 56.83      | 56.30 |
> >
> > After 100B tokens of training, the ReX model shows a modest but consistent improvement of **0.37%** in average accuracy. Importantly, we observe that ReX model achieves lower training loss compared to the base MoE after ~86B tokens. This trend also suggests that the performance gap is likely to widen with further training on more tokens.

---

### Official Review · Reviewer_gSw7 · 2025-11-01

**Soundness:** 3
**Presentation:** 2
**Contribution:** 3
**Rating:** 4
**Confidence:** 2

**Summary:**

This paper proposes REXMOE that enables cross-layer expert reuse to improve routing flexibility without inflating model parameters. Unlike conventional MoE designs where each layer maintains its own isolated expert pool, REXMOE allows routers to select from experts across $r$ adjacent layers, effectively expanding the candidate pool from $N$ to $rN$ experts while adding only negligible router parameters. To stabilize training with enlarged expert pools, the authors introduce a Progressive Scaling Routing (PSR) strategy that gradually increases the number of available experts during training. Experiments on models ranging from 0.5B to 7B parameters demonstrate consistent improvements in both language modeling perplexity and downstream task accuracy compared to vanilla MoE baselines, with particularly notable gains on reasoning-intensive benchmarks. The work provides a new architectural degree of freedom for designing parameter-efficient MoE-based language models, though practical adoption may be constrained by observed prefill latency degradation during inference.

**Strengths:**

- The paper introduces a conceptually elegant approach to MoE design by enabling cross-layer expert reuse, representing a meaningful departure from conventional layer-local routing. While parameter sharing exists in prior work, applying it specifically to MoE blocks with Progressive Scaling Routing offers a fresh perspective on balancing expert capacity and routing diversity. The minimal overhead (only router parameters) and consistent improvements across multiple model scales (0.5B to 7B) demonstrate practical viability and reasonable generalizability across different architectural configurations.

- The empirical study is thorough, spanning different model sizes, architectural variants, and evaluation metrics. Results show consistent improvements across most benchmarks, particularly on reasoning tasks. Ablation studies examining reuse frequency, PSR variants, and component contributions provide useful insights, while qualitative analysis of expert activation patterns offers preliminary evidence of task-specific specialization, though deeper mechanistic understanding would strengthen these observations.

- The work tackles a relevant challenge in scaling MoE architectures: enriching expert combinations without inflating parameters or reducing expert capacity. Given industry trends toward fine-grained MoE designs (Qwen3, DeepSeek-V3), this research direction has clear practical significance.

**Weaknesses:**

- The results in Figure 2(a) reveal substantial prefill speed degradation (up to 77% slowdown for short sequences with R8 configuration), which significantly limits the practical applicability of REXMOE in latency-sensitive applications. While the authors acknowledge this issue stems from increased I/O operations due to the larger expert pool, they do not explore potential mitigation strategies or provide detailed profiling to identify the exact bottlenecks. The paper would benefit from: (1) a breakdown of where time is spent during prefill (expert loading, routing computation, actual computation), (2) analysis of whether expert caching or other optimization techniques could reduce this overhead, and (3) discussion of use cases where the prefill penalty is acceptable versus problematic. Without addressing these concerns, practitioners may hesitate to adopt REXMOE despite its modeling advantages, especially for interactive applications where prefill latency directly impacts user experience.

- The experimental evaluation focuses primarily on TopK routing with specific architectural choices (GQA, shared experts in some configurations), but does not adequately explore how REXMOE interacts with other important MoE design decisions. For example, the paper does not investigate: (1) compatibility with different routing mechanisms beyond TopK (e.g., expert choice routing, soft routing), (2) interaction with different expert capacity factors and load balancing auxiliary losses, (3) performance under different expert granularities when controlling for total parameters. Table 3 compares against external baselines with different training data and configurations, making it difficult to isolate the contribution of expert reuse. More controlled comparisons—such as taking an existing MoE architecture and applying REXMOE versus carefully matched baselines—would better demonstrate the method's generalizability and help identify where it provides the most value.

- The paper claims that REXMOE "decouples expert dimensionality from per-layer budgets," but the experimental validation of this claim is limited. All experiments use fixed architectural configurations (Table 1), and there is no systematic study exploring how different combinations of expert size and reuse frequency affect performance under constant parameter budgets. For instance, the paper does not compare: (1) a model with 64 larger experts using R4 reuse versus one with 256 smaller experts using R1 (no reuse) at the same total parameter count, (2) whether maintaining expert capacity while increasing reuse outperforms decreasing expert capacity to accommodate more local experts, or (3) optimal trade-offs between expert dimensionality, number of experts, and reuse frequency across different model scales. Without these comparisons, it remains unclear whether the performance gains primarily come from the increased combinatorial flexibility of expert routing or from other factors, and practitioners lack guidance on how to configure these architectural choices for their specific constraints.

**Questions:**

- Can you provide a more detailed analysis of the prefill latency degradation and potential mitigation strategies?

- How does REXMOE compare against alternative approaches for increasing routing flexibility under controlled conditions?

- What strategies can address expert collapse at higher reuse factors, and what are the fundamental scaling limits?

---

> ### Author Response · Authors · 2025-11-23
> **Response to Reviewer gSw7 (1/6)**
>
> We appreciate the reviewer’s thoughtful comments. Below we provide clarification regarding the reviewer's concerns.
>
> ---
>
> > W1. The results in Figure 2(a) reveal substantial prefill speed degradation (up to 77% slowdown for short sequences with R8 configuration), which significantly limits the practical applicability of REXMOE in latency-sensitive applications. While the authors acknowledge this issue stems from increased I/O operations due to the larger expert pool, they do not explore potential mitigation strategies or provide detailed profiling to identify the exact bottlenecks. The paper would benefit from: (1) a breakdown of where time is spent during prefill (expert loading, routing computation, actual computation), (2) analysis of whether expert caching or other optimization techniques could reduce this overhead,
>
> We thank the reviewer for raising this important systems-related question. While Figure 2(a) shows a noticeable _relative_ slowdown under short-sequence prefill, we emphasize that the _absolute_ latency remains very small (8 ms → 35 ms for 128 tokens), far from affecting user experience in realworld applications.
>
> Next, we provide response to the three points raised by the reviewer, respectively:
>
> **(1) Breakdown of time spent during prefill**
>
> The primary difference between ReXMoE and a standard MoE during prefill is the **size of the candidate expert pool**. A larger reuse factor increases the number of *candidate* experts evaluated by the router in the prefill stage. However:
>
> - The **number of activated experts per token** remains unchanged.
> - The **expert computation kernels** are identical, and the decode path is completely unaffected.
> - Therefore, the profiling structure mirrors that of standard MoE, with the additional overhead concentrated almost exclusively in **router-side I/O and expert metadata access**.
>
> Since compute logic and communication volume remain unchanged, the overall execution closely matches that of standard MoE, and the absolute overhead stays small in practice. While we agree that profiling breakdowns are generally useful, in this specific case the value of a fine-grained decomposition is limited. For this reason, we believe a dedicated, ReXMoE-specific profiling breakdown would not substantially change this conclusion.
>
>  **(2) Potential mitigation strategies**
>
> The reviewer suggests exploring techniques such as **expert caching**. We would like to clarify the following:
>
> - To our understanding, **expert caching**[1] is mainly proposed for **heterogeneous MoE deployments** (e.g., hierarchical memory systems where “hot experts” are cached closer to compute devices). ReXMoE does not introduce new heterogeneity or memory tiers.
> - **Existing system optimizations** like Grouped-GEMM, kernel fusion, and compiler-level scheduling improvements can be **applied without modification**.
> - Because the decode stage is unchanged and dominates end-to-end inference latency, the marginal prefill overhead doesn’t justify adding extra complex optimizations.
>
>
> [1] Edge Caching of Mixture-of-Experts for Distributed Inference. arXiv preprint arXiv:2507.06567, 2025.

---

> ### Author Response · Authors · 2025-11-23
> **Response to Reviewer gSw7 (2/6)**
>
> > and (3) discussion of use cases where the prefill penalty is acceptable versus problematic. Without addressing these concerns, practitioners may hesitate to adopt REXMOE despite its modeling advantages, especially for interactive applications where prefill latency directly impacts user experience.
>
> **(3) Practical impact of prefill latency**
>
> To evaluate where the prefill penalty is acceptable versus problematic, we conducted additional profiling with significantly longer sequence lengths:
>
> | #Tokens    | MoE-2.3BA0.3B Latency (in ms) | MoE-2.3BA0.3B Throughput (in TGS) | ReX-2.3BA0.3B-R2    | ReX-2.3BA0.3B-R4     | ReX-2.3BA0.3B-R8    |
> | ---------- | ----------------------------- | --------------------------------- | ------------------- | -------------------- | ------------------- |
> | 4,096.00   | 13.9                          | 293,868.89                        | 281,857.83 (95.91%) | 271,765.00 (92.48%)  | 262,995.96 (89.49%) |
> | 8,192.00   | 21.7                          | 377,950.64                        | 375,107.13 (99.25%) | 359,752.24 (95.18%)  | 361,892.12 (95.75%) |
> | 16,384.00  | 38.9                          | 420,861.96                        | 413,553.59 (98.26%) | 411,431.91 (97.76%)  | 400,309.30 (95.12%) |
> | 32,768.00  | 80.3                          | 408,305.24                        | 396,730.11 (97.17%) | 405,821.25 (99.39%)  | 395,358.85 (96.83%) |
> | 65,536.00  | 162.6                         | 403,107.10                        | 383,837.12 (95.22%) | 403,486.21 (100.09%) | 392,812.21 (97.45%) |
> | 131,072.00 | 340.7                         | 384,763.87                        | 367,698.74 (95.56%) | 379,278.06 (98.57%)  | 366,035.86 (95.13%) |
>
> As shown above, even at sequence lengths up to 128k tokens, the maximum slowdown remains within **5%**, corresponding to an increase from **340.7 ms to 358 ms**. This overhead is **negligible in realistic settings** where prefill occurs only once and decode dominates the total latency.

---

> ### Author Response · Authors · 2025-11-23
> **Response to Reviewer gSw7 (3/6)**
>
> > W2. The experimental evaluation focuses primarily on TopK routing with specific architectural choices (GQA, shared experts in some configurations), but does not adequately explore how REXMOE interacts with other important MoE design decisions. For example, the paper does not investigate: (1) compatibility with different routing mechanisms beyond TopK (e.g., expert choice routing, soft routing), (2) interaction with different expert capacity factors and load balancing auxiliary losses, (3) performance under different expert granularities when controlling for total parameters.
>
> We appreciate the reviewer’s concerns and would like to clarify the rationale behind our experimental design choices.
>
> **(1) & (2) Routing mechanisms and load balancing**
>
> Our primary focus on Top-K routing stems from its extensive validation at large scale. Among existing MoE architectures, Top-K routing is the only family with widely reproduced results across large model and dataset scales (e.g., Switch-Transformer, Mixtral, Qwen, DeepSeek-V3). In contrast, alternative routing paradigms have not yet demonstrated stable, scalable behavior at comparable scales. Running a full training cycle for each routing variant would require substantial computational resources, making such exploration currently infeasible.
>
> Importantly, our method is **routing-agnostic**. Since ReXMoE only expands the candidate expert pool while keeping the token-to-expert selection mechanism unchanged, it is naturally compatible with any routing strategy. Exploring more routing mechanisms is an interesting direction for future work once their large-scale stability is better established.
>
> Regarding load balancing, our main experiments adopt the sequence auxiliary loss together with a bias term in the gate (similar to DeepSeekV3), which yields the strongest performance in our setting. To further address the reviewer’s concern, we additionally report results using the classic auxiliary loss:
>
> | Model                | ARC-E | HellaSwag | LAMBADA | LogiQA | OpenbookQA | PiQA  | SciQ  | SiQA  | WinoGrande | Avg.      |
> | -------------------- | ----- | --------- | ------- | ------ | ---------- | ----- | ----- | ----- | ---------- | --------- |
> | MoE-2.3B A0.3B       | 58.42 | 47.14     | 37.55   | 27.19  | 34.80      | 69.21 | 75.80 | 38.69 | 53.51      | 49.15     |
> | MoE-2.3BA0.3B-aux    | 58.08 | 46.56     | 38.77   | 27.65  | 34.20      | 68.99 | 75.30 | 38.08 | 53.51      | 49.02     |
> | ReX-2.3BA0.3B-aux-R2 | 59.94 | 47.19     | 38.73   | 27.34  | 36.40      | 69.64 | 76.60 | 39.36 | 53.83      | **49.89** |
>
> Under this classic auxiliary loss, our R2 configuration outperforms the MoE baseline by **0.81%**, demonstrating that REXMoE remains effective even when paired with a weaker regularization strategy. Notably, the auxiliary-loss MoE baseline is weaker than the main paper’s baseline (MoE-2.3B A0.3B), which means the gains we reported are achieved under a stronger baseline.
>
> Regarding the expert capacity factor: this hyperparameter is fundamentally a trade-off between **system efficiency** (throughput, communication/memory overhead) and **model quality** (fewer dropped tokens and less underfitting). Since varying the capacity factor primarily reflects systems & model performance trade-off, rather than illuminating the core architectural contribution of ReXMoE, we did not treat it as a primary axis of experiments.
>
> **(3) Expert granularity.**
>
> We have evaluated multiple expert granularities (8, 32, and 64 experts), as presented in Table 1 and Table 8. Exploring additional granularities would incur substantial computational cost and is orthogonal to our central research focus: improving MoE performance **without reducing per-expert capacity or increasing total parameters**. Therefore, we consider our current granularity study sufficient for the scope of this work.
>
> **Additional note on computational constraints**
>
> A full 100B-token pre-training run for our model variants requires approximately **1,200–2,000 GPU hours** per experiment. Given these costs, we believe our experimental design strikes a reasonable balance between breadth and depth. While alternative routing mechanisms, auxiliary losses, and capacity-factor choices have been explored in prior MoE literature, they are largely **orthogonal** to the architectural contribution of ReXMoE. Thoroughly evaluating all such combinations at scale would require substantial additional compute yet would not yield insights directly relevant to evaluating the core mechanism introduced in this paper.

---

> ### Author Response · Authors · 2025-11-23
> **Response to Reviewer gSw7 (4/6)**
>
> > W3. Table 3 compares against external baselines with different training data and configurations, making it difficult to isolate the contribution of expert reuse. More controlled comparisons—such as taking an existing MoE architecture and applying REXMOE versus carefully matched baselines—would better demonstrate the method's generalizability and help identify where it provides the most value.
>
> We would first like to clarify the rationale behind selecting the models included in Table 3. Our goal was to compare REXMoE with open-source models that:
> - have **similar parameter or activation count**,
> - are trained on **comparable amounts of data**,
> in order to provide a meaningful reference for the scaling potential of ReXMoE when scaled to larger models and larger datasets. Although these external baselines differ in data and configurations, they provide useful context for understanding ReXMoE’s scalability. Conducting fully controlled comparisons at such scale is currently infeasible due to the high cost of training a model from scratch.

---

> ### Author Response · Authors · 2025-11-23
> **Response to Reviewer gSw7 (5/6)**
>
> > W4. The paper claims that REXMOE "decouples expert dimensionality from per-layer budgets," but the experimental validation of this claim is limited. All experiments use fixed architectural configurations (Table 1), and there is no systematic study exploring how different combinations of expert size and reuse frequency affect performance under constant parameter budgets. For instance, the paper does not compare: (1) a model with 64 larger experts using R4 reuse versus one with 256 smaller experts using R1 (no reuse) at the same total parameter count, (2) whether maintaining expert capacity while increasing reuse outperforms decreasing expert capacity to accommodate more local experts, or (3) optimal trade-offs between expert dimensionality, number of experts, and reuse frequency across different model scales. Without these comparisons, it remains unclear whether the performance gains primarily come from the increased combinatorial flexibility of expert routing or from other factors, and practitioners lack guidance on how to configure these architectural choices for their specific constraints.
>
> Regarding the claim that REXMoE “decouples expert dimensionality from per-layer budgets,” we appreciate the reviewer’s request for more empirical evidence, and we provide additional clarification and new experimental results.
>
> **Comparisons involving finer-grained experts (addressing points (1) & (2))**
>
> We includ new results with finer-grained experts (256 experts), which supplement the original experiments:
>
> | Model               | ARC-E | HellaSwag | LAMBADA | LogiQA | OpenbookQA | PiQA  | SciQ  | SiQA  | WinoGrande | Avg.  |
> | ------------------- | ----- | --------- | ------- | ------ | ---------- | ----- | ----- | ----- | ---------- | ----- |
> | MoE-2.3B A0.3B-256E | 61.62 | 48.29     | 38.38   | 26.15  | 34.10      | 68.68 | 74.50 | 37.96 | 52.49      | 49.13 |
> | ReX-2.3B A0.3B-R4   | 60.94 | 47.96     | 38.75   | 28.42  | 37.00      | 70.18 | 76.30 | 39.36 | 53.12      | 50.23 |
>
> Although the 256-expert model increases the number of local experts by reducing per-expert capacity, it does not match the performance of ReXMoE with R4 reuse. This indicates that the improvements observed with ReXMoE are not merely due to a larger expert pool, but stem from the ability to expand the structural capacity **while preserving per-expert dimensionality**. This directly supports our claim that REXMoE decouples expert dimensionality from per-layer budgets.
>
> **Optimal trade-offs between expert dimensionality, number of experts, and reuse frequency (addressing point (3))**
>
> Fully characterizing the joint space of expert dimensionality, number of experts, and reuse frequency across model scales remains an open challenge. Even recent work on MoE scaling laws [1, 2] provides only partial guidance, and conducting the required large-scale sweeps is beyond our computational budget.
>
> Nonetheless, our current results offer several concrete insights:
>
> - Increasing reuse while **keeping per-expert capacity fixed** gives consistently better performance than decreasing expert capacity to increase the number of local experts.
> - Moderate reuse factors (e.g., R2–R4) provide strong gains without increasing the total parameters or altering layer budgets.
> - For practitioners operating under strict parameter constraints, maintaining sufficiently large experts while expanding the effective expert pool via reuse is a more favorable strategy than shrinking experts to create many small local experts.
>
> These observations outline practical heuristics and show that REXMoE provides a principled way to increase structural expressiveness without sacrificing per-expert dimensionality.
>
> [1] Krajewski J, Ludziejewski J, Adamczewski K, et al. Scaling laws for fine-grained mixture of experts[J]. arXiv preprint arXiv:2402.07871, 2024.
> [2] Tian C, Chen K, Liu J, et al. Towards greater leverage: Scaling laws for efficient mixture-of-experts language models[J].  arXiv preprint arXiv:2507.17702, 2025.

---

> ### Author Response · Authors · 2025-11-23
> **Response to Reviewer gSw7 (6/6)**
>
> > Can you provide a more detailed analysis of the prefill latency degradation and potential mitigation strategies?
>
> Please see response to W1.
>
> > How does REXMOE compare against alternative approaches for increasing routing flexibility under controlled conditions?
>
> Please see response to W4.
>
> > What strategies can address expert collapse at higher reuse factors, and what are the fundamental scaling limits?
>
> **Mitigating expert collapse**
>
> Thank you for the insightful question. In ReXMoE, the proposed Progressive Subset Routing (PSR) is designed specifically to alleviate early-stage expert collapse, which becomes more likely as the reuse factor increases.
>
> PSR temporarily restricts the active candidate expert pool during the initial phase of training. This curriculum has two stabilizing effects: (i) it reduces cross-layer gradient coupling introduced by reuse, and (ii) it softens reuse-induced routing sparsity, thereby lowering the chance that a few “hot” experts dominate the routing. Although PSR may slightly slow convergence (as observed in Figure 4), it allows the router to form more robust routing patterns before exposure to the full expert pool. Concretely, by gating only a subset of experts early on, the router avoids prematurely falling into choosing a small group of experts. As training progresses, PSR gradually expands the candidate pool to the full candidate pool, enabling the model to benefit from the increased structural capacity brought by expert reuse. In addition, standard MoE load-balancing techniques (e.g., load balancing losses) remain fully compatible with ReXMoE and can be applied on top of PSR to further reduce collapse risk.
>
>
> **Fundamental scaling limits**
>
> Increasing the reuse factor enlarges the candidate expert pool, which increases structural capacity and the combinatorial diversity of expert compositions. However, this also intensifies routing sparsity and amplifies imbalance: a larger pool becomes more susceptible to “hot-expert’’ effects, and training can destabilize if these imbalances are not sufficiently controlled.
>
> In our reported results, we only observe notable imbalance when the reuse factor reaches 8 or higher, corresponding to an expert pool of 512 experts. At this scale, optimizing load balance is already highly challenging even for existing MoE techniques, and the difficulty is not specific to ReXMoE. Since our work focuses on improving model capacity *without increasing parameter count*,  these configurations effectively represent the upper end of our practically manageable expert-pool size. We hope that our findings highlight the importance of studying MoE behavior in this large-expert regime and inspire future work in addressing the load imbalance challenge for ultra-large expert pools.
>
>
> ---
>
> We hope the above analysis fully addresses the reviewer’s concerns. We sincerely appreciate the reviewer’s constructive feedback, which has helped us strengthen the clarity and technical depth of the paper. We hope that our responses and revisions will merit a positive re-evaluation of our work.

---

> > ### Comment · Reviewer_gSw7 · 2025-11-27
> > **Thank you for the detailed response and new experiments**
> >
> > I thank the authors for their detailed response and the significant effort put into running additional experiments during the rebuttal period.
> >
> > I have reviewed the new results, and they address several of my concerns. Before I finalize my assessment, I have one clarifying question regarding this new baseline:
> >
> > - **Regarding `MoE-2.3B A0.3B-256E`** Could you explicitly provide the hyper-parameters for this model, specifically the `intermediate_size` (FFN hidden dimension) per expert? I want to verify that the total parameter count of this 256-expert baseline is strictly aligned with the `ReX-2.3B A0.3B-R4` model. Since `ReX-R4` uses 64 experts (reused), a standard MoE with 256 experts would need to reduce the FFN dimension by approximately a factor of 4 to maintain the same total parameter budget. Confirmation of this setting is important to ensure the fairness of the comparison.
> >
> > I look forward to your clarification.

---

> > > ### Author Response · Authors · 2025-11-27
> > >
> > > We thank the reviewer for the response and follow-up discussion.
> > >
> > > For `MoE-2.3B A0.3B-256E`, the `intermediate_size` is set to `186`, and the number of activated experts is set to `32`, so that the total number of activated parameters matches that of `ReX-2.3BA0.3B-R4`.
> > >
> > > For reference, both `ReX-2.3B A0.3B-R4` and its base model `MoE-2.3BA0.3B` use `intermediate_size = 744` per expert, which is exactly four times 186.
> > >
> > > We hope this clarifies the configuration and are happy to discuss any additional questions or concerns.

---

### Author Response · Authors · 2025-12-04
**Summary of the Discussion Phase**

We thank the reviewers for their valuable feedback and discussion. The clarity and technical depth of the our work has been significantly strengthened during this phase, and the key improvements are summarized below:

---
## **Method Clarity Improvements** (`xYSR`, `7tRi`)

We provide theoretical analysis for the two key components of ReXMoE. Expert reuse expands the effective expert pool by a factor of  $\frac{C(rN, k)}{C(N, k)} = \prod_{i=0}^{k-1} \frac{rN - i}{N - i}$, thereby increasing the combinatorial space available to the router. However, reuse introduces **gradient coupling**, where PSR is designed to mitigate this issue by alleviating the coupling in early-stage training and gradually expand the candidate pool to fully utilize the structural capacity.

---
## **Practical Impact on System Behavior**

### **Inference Speed Under Short Sequences** (`gSw7`)
The short-sequence latency increase (8 ms → 35 ms) is negligible. At other lengths, ReXMoE matches standard MoE performance.

### **Impact on Longer Sequences and Larger Models** (`gpR3`, `zRRq`)
We provide additional evaluations up to 128k sequence length and on larger models (7BA1B and 30BA1B). ReXMoE shows **no meaningful inference degradation** on these settings.

### **Impact on Parallel Strategies** (`xYSR`, `7tRi`, `zRRq`)
We analyze the impact of common parallel strategies such as Expert Parallelism (EP) and Pipeline Parallelism (PP). With an appropriate PP–EP configuration, ReXMoE introduces **no additional communication overhead** and does not create new scaling bottlenecks for either training or inference.

---
## **Additional Experimental Results**

### **Significance of the Improvement** (`xYSR`, `7tRi`, `gpR3`)
We expanded the 2.3B base MoE to 3.5B (+50% params) for only a 0.77% gain, while ReX-R4 delivers larger improvements without adding parameters.

### **Applying PSR to Vanilla MoE** (`xYSR`)
We apply PSR to vanilla MoE and observe a **0.77% degradation**. This is expected: gradient coupling does not exist without expert reuse.

### **Training Under Larger Budget** (`zRRq`)
We trained a MoE-155MA46M model on 350B tokens. The ReX model improves by **0.92%**, showing stability under larger budgets.

### **Training on Larger Models** (`xYSR`, `7tRi`)
For a 7BA1B configuration, the ReX-R4 model achieves a **0.37%** improvement, confirming benefits of ReXMoE at larger scales.

### **Using Classic Auxiliary Load-Balance Loss** (`gSw7`)
Under the classic auxiliary balancing loss, our R2 configuration outperforms the MoE baseline by **0.81%**, confirming that ReXMoE is compatible with standard load-balancing mechanisms.

### **Comparison with Finer-Grained Experts** (`gSw7`)
We evaluate models using experts with $1/4$ the original dimension while keeping total and active parameters fixed. ReXMoE outperforms this finer-grained setting by **1.1%**, showing that reuse enhances performance without compromising expert capacity.

---
## **Additional Discussion & Analysis**

### **Expert Collapse & Mitigating Load Imbalance** (`gSw7`, `gpR3`, `7tRi`)
We further analyze the causes of expert collapse and load imbalance under large reuse factors. Large reuse factors (>8) push the effective pool beyond 512 experts, a challenging regime for current balancing methods. ReXMoE remains compatible with standard MoE balancing, and the enlarged pool behaves like a conventional larger MoE.
### **Hyperparameter Tuning for PSR** (`gpR3`)
PSR uses only two hyperparameters ($t_s, t_e$). It mainly operates mid-training to mitigate coupling early and enable full reuse later, keeping tuning costs low.
### **Generalization to Other MoE Architectures** (`gpR3`)
ReXMoE can be applied to any MoE design. It expands the candidate pool without altering backbone, routing, or balancing methods, and complements other MoE improvements.
### **Task Specialization and Benchmark Trends** (`gpR3`)
- Additional visualizations (Fig. 10) reveal balanced mean activation ratios and task-specific activation patterns.
- We attribute benchmark-specific accuracy trends to domain mismatch between tasks and our training corpus.
### **Clarification on Comparison with Open-Source Models** (`gSw7`, `zRRq`)
Table 3 compares open-source models with similar activation budgets and training scales, providing meaningful context for assessing ReXMoE’s scaling potential.
### **Clarification on Experimental Design** (`gSw7`)
Our experiments focus on demonstrating that ReXMoE improves performance without increasing expert parameters or weakening expert capacity. Fully exploring the joint design space (expert size/count, routing, losses, reuse frequency, etc.) is computationally prohibitive. Current results (different architectures and scales, training budgets, load-balancing losses) sufficiently support our claims.

---
## **Minor Fixes**
We fixed several typos (`gpR3`, `zRRq`) and added the computation of activation ratio (`gpR3`).

---

We have included above results and fixes in our revised version.

---

### Author Response · Authors · 2025-12-04
**Response Summary for the Area Chair**

We thank the new AC for stepping in at this challenging stage of the review process and for taking over the handling our submission. We sincerely appreciate your time and effort. Below, we summarize the main discussion points raised during the rebuttal and the improvements we have made for your consideration.

Overall, the reviewers’ comments focused on the following aspects:

- **Theoretical analysis of the method** (`xYSR`, `7tRi`)
- **Additional experimental results to support our claims** (`gSw7`, `gpR3`, `xYSR`, `7tRi`, `zRRq`)
- **Further discussion and analysis on several technical issues** (`gSw7`, `gpR3`, `xYSR`, `7tRi`, `zRRq`)
- **Minor issues, such as typos** (`gpR3`, `zRRq`)

We believe that our rebuttal and subsequent responses have **fully addressed all these concerns**, and a detailed breakdown is provided in the summary post below.

We would also like to reiterate the **positioning and core contribution** of our work. Our main innovation lies in exploring how to achieve enriched combinatorial space **without increasing the parameter budget or reducing the representational capacity of each expert** by reusing experts across layers. Progressive scaling routing (PSR) is then introduced to effectively realize the enlarged structural capacity by mitigating early-stage gradient coupling. Our method requires only modest additional router parameters to enable cross-layer expert selection. It is **orthogonal to existing MoE designs** and can be incorporated directly into current MoE architectures.

We conducted dozens of **100B-token, from-scratch training runs** across different scales and architectural configurations. The results consistently show that ReXMoE delivers strong improvements, providing solid empirical support for both our methodology and our claims. We have also carefully examined the practical deployment feasibility of the proposed approach, demonstrating that it introduces **no additional system overhead**.

We respectfully encourage the AC to consider the **parameter-efficient, architecture-orthogonal, and practical** nature of our contributions, which we believe open a promising new paradigm for designing MoEs.

We kindly ask the AC to review our revised paper, rebuttal, and follow-up responses (including additional experiments). We hope that our responses and revisions will merit a positive evaluation of our work. We sincerely appreciate your time, attention, and efforts in handling our submission.

---

### Meta-Review · Area_Chair_UUht · 2026-01-07

**Summary:**

The initial reviews of this submission were overall negative. Several critical concerns were raised regarding its practical utility and theoretical grounding:
1. Inference Latency: Reviewers (gSw7, gpR3, zRRq) noted significant prefill latency degradation (up to 77% for short sequences) and questioned the scalability of the large expert pool.
2.Ablation and Motivation: Reviewers (xYSR, 7tRi) questioned the necessity of expert reuse versus the Progressive Scaling Routing (PSR) strategy, suggesting PSR might be the primary driver of gains.
3. System Scalability: Concerns were raised regarding how cross-layer reuse impacts standard Pipeline Parallelism (PP) and Expert Parallelism (EP) strategies (xYSR, 7tRi, zRRq).
4. Lack of Baselines: Reviewers requested more controlled comparisons against finer-grained MoE baselines (where parameter count is matched by shrinking expert size) and evaluations at the 7B scale (gSw7, 7tRi).

**Reviewer Concerns:**

Addressed Concerns:
1. Latency Characterization: The authors provided extensive profiling across sequence lengths up to 128k tokens.
2. Theoretical & Empirical Grounding of PSR): The authors clarified that expert reuse increases combinatorial space but introduces "gradient coupling." PSR acts as a necessary curriculum to mitigate this coupling.

Outstanding Concerns:
1. Fundamental Scaling Limits: As the authors noted, when the reuse factor $r$ reaches very high numbers (e.g., 32), the candidate pool exceeds 2000 experts, leading to "hot-expert" collapse. While PSR mitigates this, the fundamental limits of load balancing for ultra-large pools remain a research frontier.
2. Benchmark Performance: While REXMOE shows clear relative gains over vanilla MoE baselines, its absolute performance on specialized reasoning tasks (e.g., GSM8K) remains modest when trained on general web data, which is more a function of the training corpus than the architecture.

**Reviewer Scores:**

gSw7	4 ->4/6	Addressed latency profiling and finer-grained baselines.
xYSR	6 ->6	Satisfied with PP/EP scalability analysis and 7B results.
gpR3	4 ->4/6 Clarified prefill overhead and PSR hyperparameter tuning.
7tRi	4->4/6. Theoretical justification for PSR/Reuse synergy was well-received.
zRRq	4 ->4/6 Long-context inference benchmarks.

---

### Decision · Program_Chairs · 2026-01-26

Reject